# Antarctic permafrost processes and antiphase dynamics of cold-based glaciers in the McMurdo Dry Valleys inferred from [10]Be and [26]Al cosmogenic-nuclides

Jacob T.H. Anderson[1], Toshiyuki Fujioka[2], David Fink[3], Alan J. Hidy[4], Gary S. Wilson[1,5], Klaus Wilcken[3], Andrey Abramov[6], Nikita Demidov[7]

[1]Department of Marine Science, University of Otago, PO Box 56, Dunedin, New Zealand

[2]Centro Nacional de Investigación sobre la Evolución Humana, Burgos 09002, Spain

[3]Australian Nuclear Science & Technology Organisation, New Illawarra Road, Lucas Heights, NSW, 2234, Australia

[4]Center for Accelerator Mass Spectrometry, Lawrence Livermore National Laboratory, Livermore CA 94550, USA

[5]GNS Science, PO Box 30368, Lower Hutt 5040, Wellington, New Zealand

[6]Institute of Physicochemical and Biological Problems of Soil Science, Pushchino, Russia

[7]Arctic and Antarctic Research Institute, St. Petersburg, Russia

*Correspondence to:* Jacob T.H. Anderson (jacob.anderson@otago.ac.nz)

## Abstract

Soil and sediment mixing and associated permafrost processes are not widely studied or understood in the McMurdo Dry Valleys of Antarctica. In this study, we investigate the stability and depositional history of near-surface permafrost sediments to ~3 m depth in Pearse and lower Wright valleys using measured cosmogenic [10]Be and [26]Al depth profiles. At Pearse Valley, we estimate a minimum depositional age of ~74 ka for the active-layer and paleoactive-layer sediments (<0.65 m). Combined depth profile modelling of [10]Be and [26]Al gives a depositional age for near-surface (<1.65 m) permafrost at Pearse Valley of $180 \, ^{+20}/_{-40}$ ka, implying deposition of permafrost sediments predate MIS 5 advances of Taylor Glacier. Deeper permafrost sediments (>2.09 m) at Pearse Valley are thus inferred to have a depositional age of >180 ka. At a coastal, lower elevation site in neighbouring lower Wright Valley, [10]Be and [26]Al depth profiles from a second permafrost core exhibit near-constant concentrations with depth, and indicate the sediments are either vertically mixed after deposition, or are sufficiently young and post-depositional nuclide production is negligible relative to inheritance. [26]Al/[10]Be concentration ratios for both depth profiles range between 4.0 and 5.2 and are all lower than the nominal surface production rate ratio of 6.75, indicating that prior to deposition, these sediments experienced complex,

yet similar, exposure-burial histories. Assuming a single cycle exposure-burial scenario, the observed
$^{26}Al/^{10}Be$ ratios are equivalent to a total minimum exposure-burial history of ~1.2 Ma.
In proximity to the depth profile core site, we measured cosmogenic $^{10}Be$ and $^{26}Al$ in three granite
cobbles from thin, patchy drift (Taylor 2 Drift) in Pearse Valley to constrain the timing of retreat of
Taylor Glacier. Assuming simple continuous exposure, our minimum, zero erosion, exposure ages
suggest Taylor Glacier partially retreated from Pearse Valley no later than 65–74 ka. Timing of retreat
after 65 ka and until the Last Glacial Maximum (LGM) when Taylor Glacier was at a minimum position,
remains unresolved. The surface cobble ages and permafrost processes reveal Taylor Glacier advances
during MIS 5 were non-erosive or mildly erosive, preserving the underlying permafrost sediments and
peppering boulders and cobbles upon an older, relict surface. Our results are consistent with U/Th ages
from central Taylor Valley, and suggest changes in moisture delivery over Taylor Dome during MIS
5e, 5c and 5a appear to be associated with the extent of the Ross Ice Shelf and sea ice in the Ross Sea.
These data provide further evidence of antiphase behaviour through retreat of a peripheral lobe of
Taylor Glacier in Pearse Valley, a region that was glaciated during MIS 5. We suggest a causal
relationship of cold-based glacier advance and retreat that is controlled by an increase in moisture
availability during retreat of sea ice and perhaps the Ross Ice Shelf, and conversely, a decrease during
times of sea ice and Ross Ice Shelf expansion in the Ross Sea.

## 1 Introduction

Permafrost (perennially frozen ground) in the McMurdo Dry Valleys, Antarctica, contains valuable
records of paleoenvironmental information, yet the stability of permafrost sediments, and the processes
that influence sediment transport, erosion and deposition in the McMurdo Dry Valleys are not well
understood. Previous studies investigating chronology and stability of glacial drift deposits, sediments
and permafrost in the McMurdo Dry Valleys and Transantarctic Mountains typically focused on high
elevation sites (e.g., Bergelin et al., 2022; Bibby et al., 2016; Morgan et al., 2011; 2010; Ng et al., 2005;
Schäfer et al., 2000; Sugden et al., 1995). The objective of these studies has largely been to constrain
the ages and / or erosion and sublimation rates of early Pleistocene, Pliocene, and Miocene landscapes.
There only appears to be one study investigating the age and stability of permafrost below 1000 m
elevation (Morgan et al., 2010). Yet, understanding the depositional environment and stability of
permafrost at low elevations is important for interpreting landscape evolution, geomorphic processes
and polar climate change on Earth, and as a terrestrial analogue for Mars (e.g., Marchant & Head, 2007).
Studies have also revealed permafrost contain frozen reservoirs of ice, greenhouse gases, ancient
bacteria, and viruses (Adriaenssens et al., 2017; Gilichinsky et al., 2007; Ruggiero et al., 2023). Future
thawing of low elevation environments, from increasing atmospheric temperatures, could increase
microbial activity and release previously frozen gases, and nutrients, leading to unprecedented changes
in hydrological, and biogeochemical cycles.
Permafrost usually contains an active, cryoturbated, mobile sediment layer, up to ~70 cm in depth.
Active-layer thickness, thawing, and permeability is modulated by seasonal variations. Permafrost
sediments are episodically covered by advancing and retreating ice (Atkins, 2013), which can further
complicate the interpretation of permafrost stability, sediment transport and mixing. In the McMurdo
Dry Valleys, there is currently no clear trend of increase or decrease in active-layer thickness between
2006 and 2019 (Hrbáček et al., 2023). The lack of understanding permafrost dynamics limits our ability
to reconstruct permafrost stability or evolution through time. Further research is needed to explore the
rates and mechanisms by which sediments are transported and mixed via aeolian, fluvial, and periglacial
processes.
Key components influencing permafrost processes and overlying geomorphic landforms are the
climatic conditions and extent of the Antarctic ice sheets. During Plio-Pleistocene warm intervals, the
West Antarctic Ice Sheet (WAIS), and marine-based sectors of the East Antarctic Ice Sheet (EAIS)
underwent extensive retreat (Naish et al., 2009; Pollard & DeConto, 2009; Cook et al., 2013; Blackburn
et al., 2020; Patterson et al., 2014). Warmer than present global temperatures and higher than present
sea levels are also observed in recent prominent interglacial periods, i.e., MIS 31 (~1.07 Ma), MIS 11
(~400 ka), and MIS 5e (130 - 115 ka) (Dutton et al., 2015; Naish et al., 2009; Pollard & DeConto,
2009). The extent of ice sheet retreat during these recent warm intervals varied significantly within
different drainage basins and through time. During the penultimate interglacial (MIS 5e), the average
global temperature was ~1–2°C warmer than pre-industrial (Fischer et al., 2018; Otto-Bliesner et al.,
2013), Antarctic temperatures were ~3–5°C warmer (Jouzel et al., 2007) and global mean sea levels
were ~6–9 metres higher than present (Dutton & Lambeck, 2012; Kopp et al., 2009). With a global
average temperature currently ~1.1°C warmer than pre-industrial levels, and predicted to be ≥1.5°C in
the coming decades (IPCC, 2021), interglacial conditions, such as during MIS 5, are an important
analogue for evaluating future ice sheet behaviour and global climate processes under future warming
scenarios.

Simulated ice sheet retreat during MIS 5e by Golledge et al. (2021) suggested ice loss in the Thwaites
and Pine Island sector of the WAIS, whereas the Ross Ice Shelf remained intact. Conversely,
simulations by DeConto & Pollard (2016), and Turney et al. (2020) suggested retreat of the Ross Ice
Shelf, followed by retreat of the WAIS interior. The $\delta^{18}O$ ice core records from Talos Dome reveal the
EAIS was relatively intact during MIS 5 (Sutter et al., 2020) and recent studies suggest partial ice sheet
lowering in Wilkes Subglacial Basin but no grounding line retreat (Fig. 1; Golledge et al., 2021; Sutter
et al., 2020; Wilson et al., 2018). Ice core studies reveal increased accumulation rates at Taylor Dome
(Steig et al., 2000) and the Allan Hills Blue Ice Area (Yan et al., 2021) near the onset of the Last
Interglacial. Yan et al. (2021) hypothesized that high accumulation rates during warm interglacials may
reflect open ocean conditions in the Ross Sea, caused by reduced sea ice extent, and possibly retreat of
the Ross Ice Shelf relative to its present-day position. This hypothesis is supported by a depleted $\delta^{18}O$
value (−0.175 ‰) from ice core records at Roosevelt Island, indicating high sea level and reduced ice
sheets during MIS 5a (Lee et al., 2020).

In contrast, terrestrial evidence from the McMurdo Dry Valleys suggests Taylor and Ferrar glaciers
were larger than present during warm interglacials of the mid-Pliocene climatic optimum (3.0–3.1 Ma),
MIS 31 (1.07 Ma) (Swanger et al., 2011) and MIS 5 (Brook et al., 1993; Higgins et al., 2000a). These
glacier advances appear to be out of phase with WAIS retreat and ocean warming during interglacial
periods. Alpine glaciers in the McMurdo Dry Valleys also appear out of phase with marine based ice
sheet retreat and advanced during MIS11 (Swanger et al., 2017), MIS 5 (Swanger et al., 2019), and MIS
3 (Joy et al., 2017). The past ice volume and extent of Taylor Glacier (during interglacial periods) has
been derived from cosmogenic nuclide studies and mapping drift and moraine deposits in lower Kennar
Valley (Swanger et al., 2011), and lower Arena Valley (Brook et al., 1993; Marchant et al., 1994), and
U/Th dating in central Taylor Valley (Higgins et al., 2000a). MIS 5 age glacial deposits in central Taylor
Valley and Arena Valley are mapped as Taylor 2 Drift (Bockheim et al., 2008; Brook et al., 1993; Cox
et al., 2012; Denton et al., 1970), termed Bonney Drift by Higgins et al. (2000b). By inference, glacial
deposits on the valley floor of Pearse Valley are mapped as Taylor 2 Drift (Bockheim et al., 2008; Cox
et al., 2012; Denton et al., 1970). U/Th ages of algal carbonates in central Taylor Valley suggest multiple
advance / retreat cycles of the Taylor Glacier snout during MIS 5,  with retreat of Taylor Glacier
continuing after the MIS 5/4 transition (Higgins et al., 2000a). The $\delta^{18}O$ values measured from buried
ice in northern Pearse Valley also support the advance of Taylor Glacier during MIS 5 (Swanger et al.,
2019). However, the timing of advance and retreat of Taylor Glacier in central Taylor Valley and in
Pearse Valley remain poorly constrained.
In this study, we investigate the stability and depositional history of near-surface permafrost sediments
using paired $^{10}Be$ and $^{26}Al$ depth profiles of permafrost from Pearse and lower Wright valleys. We
compare the exposure-burial history of the permafrost cores from the two sites and the long-term
recycling processes of McMurdo Dry Valleys sediments. We also investigate the relationship between
thin, patchy drift overlying permafrost sediments in Pearse Valley. Thin, patchy drift is the only
evidence of cold-based glacier overriding, and is defined as a scattering of clasts overlying older,
undisturbed desert pavements (Atkins, 2013). We present cosmogenic nuclide surface exposure ages
from three cobbles in Pearse Valley to determine the age of Taylor 2 Drift, and provide constraints on
the timing of retreat of a peripheral lobe of Taylor Glacier during MIS 5. Combining permafrost depth
profiles and exposure ages of cobbles from the drift, we infer the depositional history of the permafrost

sediments and constrain a minimum age of Taylor Glacier retreat. These data from Pearse Valley provide insight into Taylor Glacier behaviour and associated geomorphic processes during MIS 5.

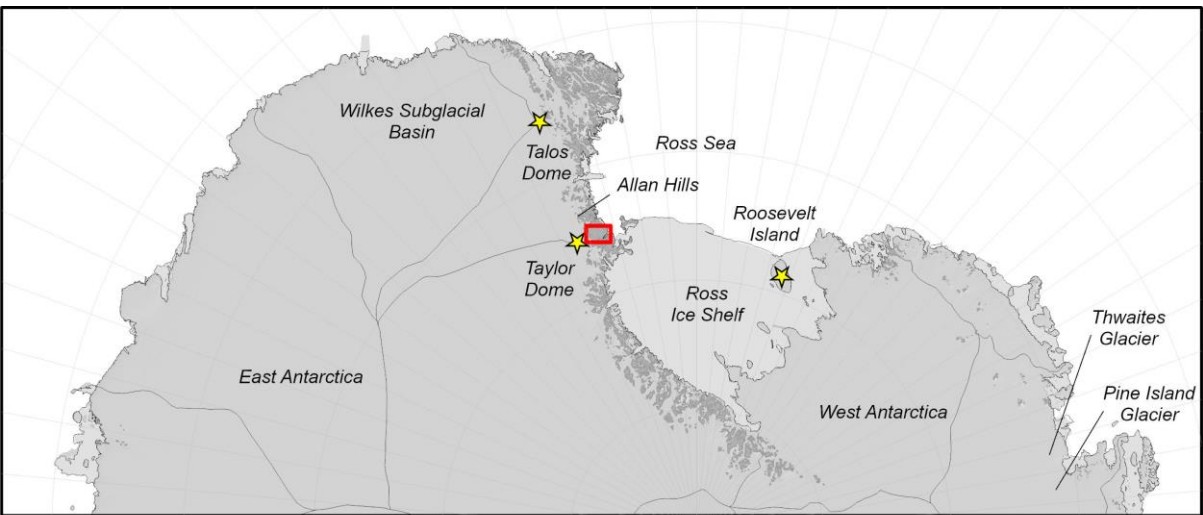

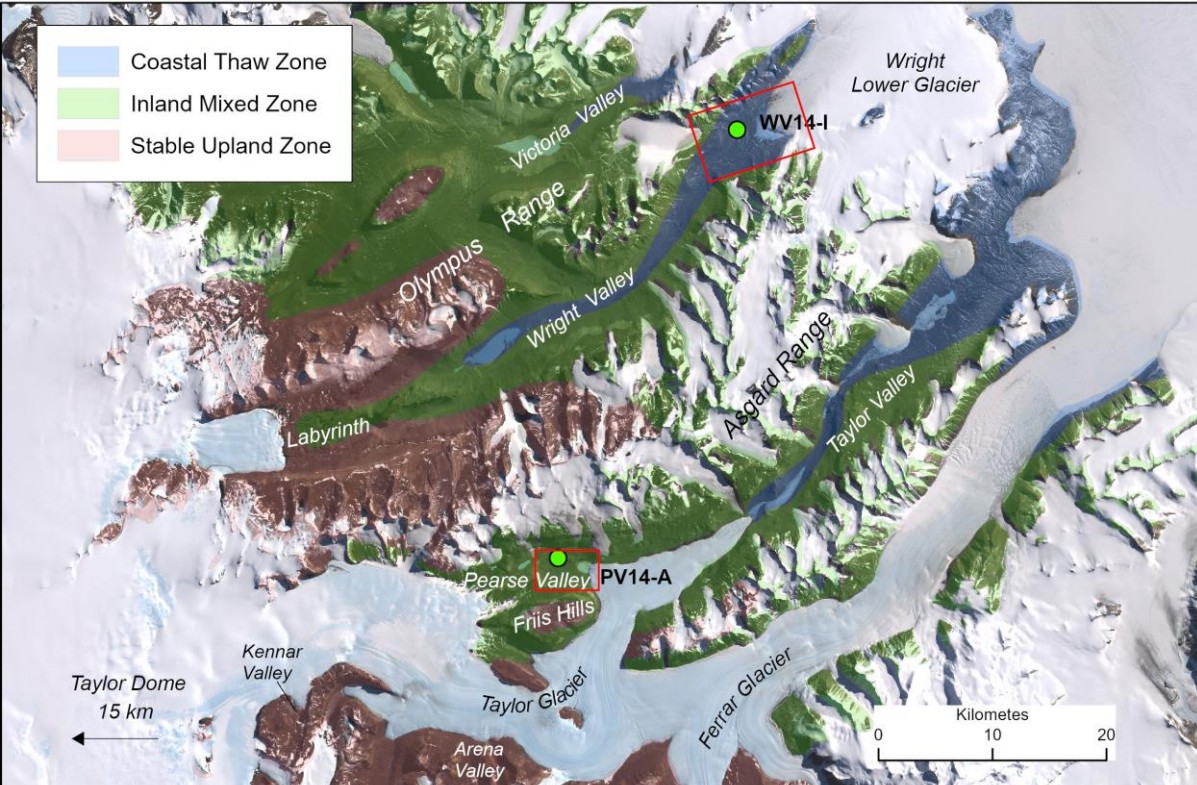

**Figure 1.** Study area and location of McMurdo Dry Valleys. Yellow stars show ice core sites discussed in the text. The green circles show the locations of the Pearse Valley and lower Wright Valley sites where permafrost cores were recovered. The three microclimatic zones are the stable upland zone (brown), inland mixed zone (green), and coastal thaw zone (blue). Modified from Marchant and Head (2007); and Salvatore and Levy, (2021). Red rectangles in the lower diagram show the locations of Pearse Valley in Fig. 2 and lower Wright Valley in Fig. 3.

## 2 Geologic setting and study area

The Dry Valleys are a hyperarid, cold polar desert and can be subdivided into three geographic zones (stable upland, inland mixed, and coastal thaw zones), which are defined by their microclimatic parameters of atmospheric temperature, soil moisture, and relative humidity (Fig. 1; Marchant & Denton, 1996; Marchant & Head, 2007). The stability and evolution of geomorphic features and permafrost are controlled by subtle variations within each microclimatic zone. The active-layer in permafrost is defined as soil horizons where the ground temperature fluctuates above and below 0°C seasonally (Davis, 2001; Yershov, 1998). Antarctic permafrost soils along the floors and flanks of ice-free valleys are vertically mixed, initially through deposition of reworked sediments, and secondarily through active-layer cryoturbation up to 70 cm depth of the surface (Bockheim et al., 2007; 2008). Cryoturbation is defined as soil movement due to repeated freeze–thaw, generally within the active-layer of permafrost (French, 2017). Active-layers can be distinguished by the presence (wet active-layer) or absence (dry active-layer) of water. Soils in the coastal thaw zone are seasonally moist and comprise wet active-layers, whereas soils in the inland mixed zone are dry and comprise dry active-layers (Marchant & Head, 2007). Our study sites focused on two different microclimatic zones (Fig. 1); Pearse Valley in the inland mixed zone, and lower Wright Valley in the coastal thaw zone, which differ in age, elevation, and distance from the coast.

### 2.1 Pearse Valley

Pearse Valley is an ice-free valley that is bounded by the Friis Hills in the south, the Asgard Range in the north and opens onto peripheral lobes of Taylor Glacier in the east and west (Fig. 1). Taylor Glacier flows east from Taylor Dome of the EAIS, terminating in Taylor Valley. At the eastern end of Pearse Valley, a lobe of Taylor Glacier terminates into Lake Joyce, a closed-basin proglacial lake (Fig. 2). Taylor Glacier and local alpine glaciers have advanced in the present interglacial and occupy their maximum position since the Last Glacial Maximum (LGM) (Higgins et al., 2000a). At the head of Pearse Valley, glacially incised bedrock sits at a similar elevation to the Labyrinth platform in upper Wright Valley, likely formed by a network of subglacial drainage channels beneath wet-based glacial conditions during the Miocene Climate Transition (Fig. 1; Lewis & Ashworth, 2016; Chorley et al., 2022). The northern valley wall comprises gelifluction lobes, buried snowpack deposits, meltwater channels derived from ephemeral streams, and fans fed by the meltwater channels in front of the lobes (Heldmann et al., 2012; Swanger et al., 2019). The valley floor consists of a lower elevation area on the southern side, and a higher elevation area on the northern side of the valley. The PV14-A core and cobble samples are located on the central northern side of the valley floor (Fig. 2).

The local bedrock comprises basement granites and Ferrar dolerite intrusives (Cox et al., 2012; Gunn & Warren, 1962). Glacial deposits on the valley floor are mapped as Taylor 2 Drift (Bockheim et al., 2008; Denton et al., 1970). These sediments were inferred as waterlain and melt-out tills following the

penultimate down-valley advance of the Taylor Glacier during MIS 5 (70 – 130 ka) (Cox et al., 2012;
Higgins et al., 2000a; Swanger et al., 2019). The valley floor landscape is characterized by hummocky
moraines with a combination of glacigenic, and fluvial deposits, and aeolian sediments. Variably
weathered granite boulders (up to 3 m in diameter) form a lag deposit on the drift surface, inferred as a
till deflation or a separate younger depositional unit (Higgins et al., 2000b). The northern and southern
Pearse Valley walls comprises extensive rock glaciers (Swanger et al., 2019).

**2.1.1 Modern climate**
Pearse Valley is situated in the inland mixed zone of the Dry Valleys (Marchant & Denton, 1996). The
valley has a mean annual temperature of -18°C (Marchant et al., 2013) and precipitation rates of 20–50
mm/yr (water equivalent), and 100–200 mm/yr in the adjacent Asgard Range, the source region for the
local alpine glaciers (Fountain et al., 2010). Mean summer air temperatures (December through
February) in Pearse Valley are -2 to -7°C (Marchant et al., 2013). Ground surface temperatures measured
at the Pearse Valley meteorological station between 27–28 November, 2009, recorded a peak
temperature of 10°C due to solar heating (Heldmann et al., 2012). Winds in Pearse Valley are strong
enough to mobilise sand grains and form aeolian surface features such as sand dunes (Heldmann et al.,
198 2012).

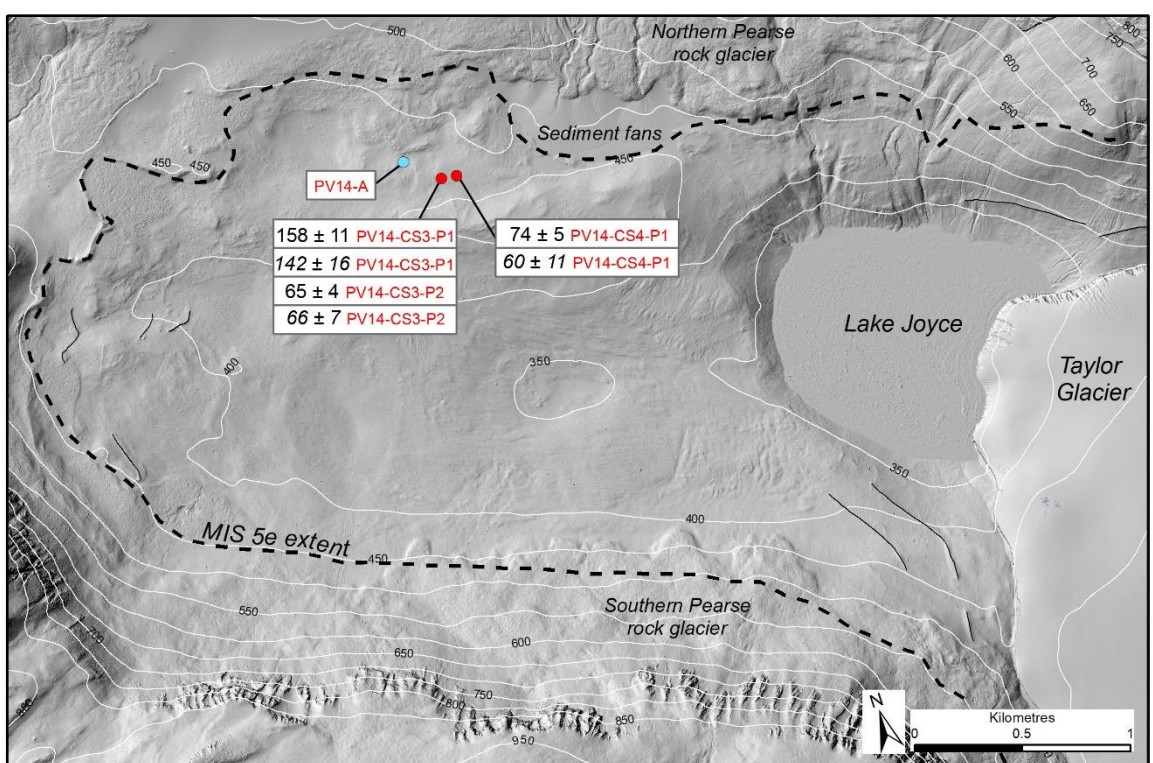


**Figure 2.** Map of Pearse Valley with MIS 5e extent of Taylor Glacier (black dashed line; Cox et al.,
2012), sample locations and PV14-A permafrost drill site (blue circle). Thin black lines trace undated
moraines. PV14-A drill site and measured [10]Be and [26]Al (italics) ages of cobbles residing on boulders
are shown in kiloyears with 1σ uncertainties (red circles). Lidar image from Fountain et al. (2017).

## 2.2 Lower Wright Valley

Lower Wright Valley is ice-free and is bounded by the Asgard Range in the south, and the Olympus
Range in the north (Fig. 1). The mouth of the valley at the eastern end is cut off from the Ross Sea by
the Wright Lower Glacier, a lobe of the Wilson Piedmont Glacier. Lake Brownworth, a proglacial lake
fed by the Wright Lower Glacier, supplies the westward flowing Onyx River. The WV14-I core is
located on the northern side of lower Wright Valley (Fig. 3). Radiocarbon dates of lacustrine algae from
glaciolacustrine deposits suggest Lake Brownworth is a small remnant of a much larger lake that existed
during the LGM and early Holocene (Hall et al., 2001). The post-glacial, Holocene age landscapes form
hummocky moraines, with a combination of deltas, shorelines and glaciolacustrine sediments (Hall et
al., 2001). Glacial meltwater streams drain into Lake Brownworth and the Onyx River from the north
and south valley walls. The local bedrock comprises basement metasediments and granites, and Ferrar
dolerite intrusives (Cox et al., 2012). Metasediments, granite, dolerite and occasional basalt sediments
in the lower Wright Valley have accumulated since the last deglaciation by lacustrine, fluvial and
aeolian processes (Hall et al., 2001; Hall & Denton, 2005).

### 2.2.1 Modern climate

Lower Wright Valley is situated in the coastal thaw zone of the McMurdo Dry Valleys (Marchant &
Denton, 1996) and has a mean annual temperature of -21°C (Doran et al., 2002) and precipitation rates
of 26–51 mm/yr (water equivalent) (Fountain et al., 2010). Mean summer air temperatures (December
through February) in lower Wright Valley are -5 to -7°C, and can exceed 0°C for >6 days per year
(Doran et al., 2002). Meltwater forms during summer months (December and January) when
temperatures can rise to as much as 10°C at some locations (Hall et al., 2001).

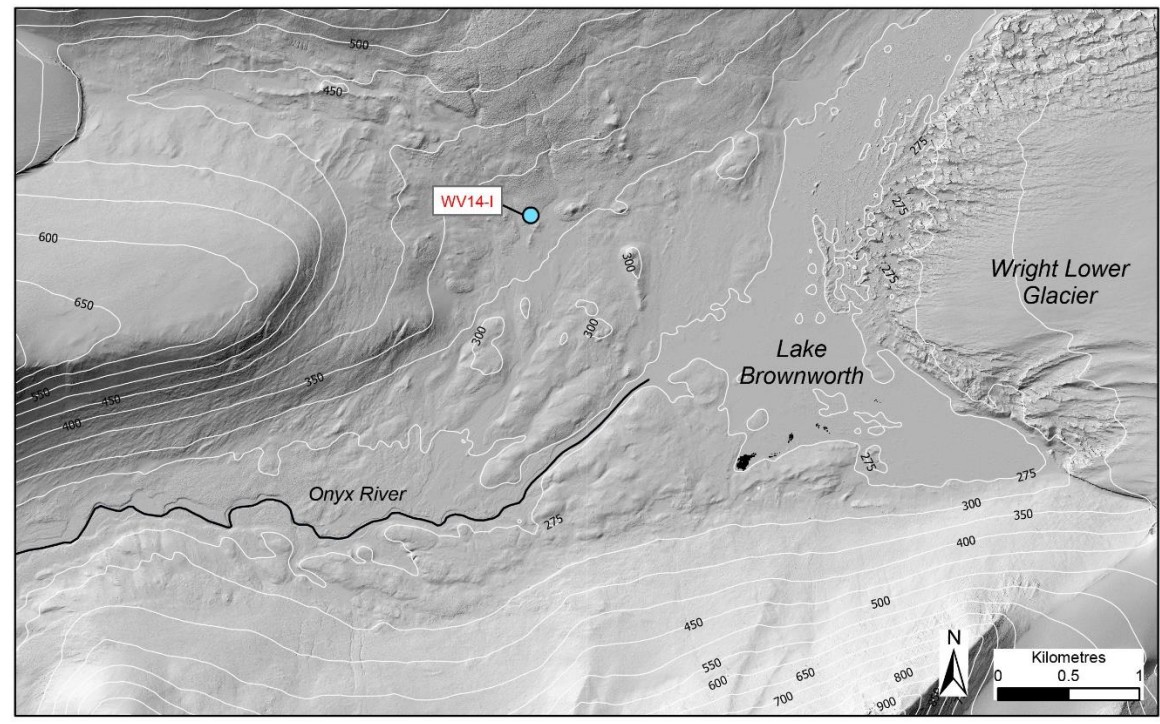


**Figure 3.** Map of lower Wright Valley and WV14-I permafrost drill site (blue circle). Lidar image from
Fountain et al. (2017).


## 3 Methods

### 3.1 Permafrost core locations and characteristics

During the 2014/15 austral field season, permafrost cores were recovered from Pearse Valley and lower
Wright Valley using a gasoline powered dry drilling technique (Fig. 1). These two cores were sampled
for sedimentological and for cosmogenic nuclide analysis. After extraction, the core sections were
divided into ~10 cm portions for sub-sampling and analysis. The upper sections were collected in Whirl-
Pak bags as the core recovery was poor. Core integrity below the active-layer in ice-cemented
permafrost sediments was good and cores were collected as rigid intact sections in PVC core liners.

239

### 3.1.1 Pearse Valley borehole core

The PV14-A core is located on an elevated bench that extends along the northern side of the valley floor
at 450 masl (77.7062°S, 161.5467°E), ~3 km north-west of the present position of the Taylor Glacier
lobe (Fig. 2). The core was recovered to a depth of 3.16 m (Fig 4). The active-layer (0 – 0.37 m) above
the ice-cemented permafrost consists of a thin armoured surface layer of desert pavement (~0.02 m
thick), and a layer of loose dry sand (~0.35 m thick). Recovered sediments from beneath the armoured
desert pavement comprise a dry active-layer of loose sand and pebbles down to 0.37 m depth. Below

0.37 m depth, the recovered sediments comprise ice-cemented permafrost, with grains of sand and pebbles forming the matrix, and the pore spaces filled with ice. The $^{10}$Be and $^{26}$Al depth profiles (Fig. 4) start below the 0.02 m thick surface armoured pavement. The first three samples were collected from the dry active-layer followed by nine from the ice-cemented permafrost. Sediments within the permafrost core comprise gravelly sands derived from weathered Beacon Supergroup, granite, granodiorite, diorite, and dolerite origins. They appear structureless, or weakly bedded which we interpret to be fluvio-glacial and aeolian deposits. Between 0.73–0.86 m depth, the core comprises several ice lenses indicative of ice accumulation below a paleosublimation unconformity. Several small ice lenses were also recovered between 1.57–1.87 m depth. The ice lenses are typically clean ice or debris-poor ice compared to adjacent upper and lower segments. Only two of the three active-layer samples, and six of the nine permafrost core samples were successful in providing paired $^{10}$Be and $^{26}$Al concentrations (Fig. 4; Table 1).

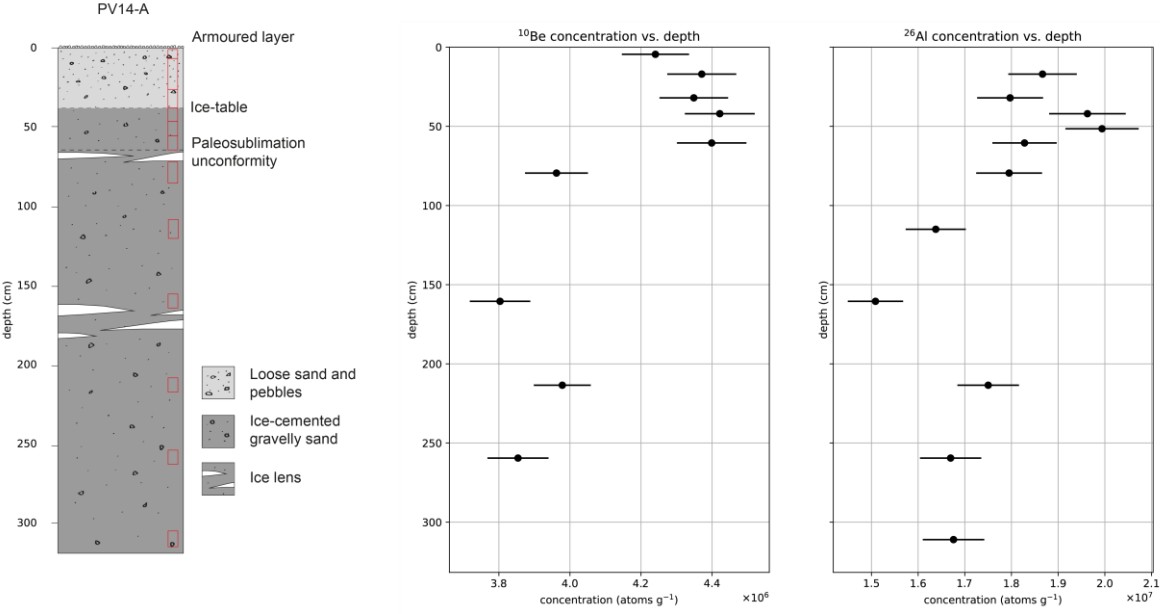

**Figure 4.** Pearse Valley (PV14-A) permafrost core sedimentology (left). Locations of cosmogenic nuclide samples shown in red boxes. The modern active-layer is from 0–0.37 m depth. Pearse Valley (PV14-A) permafrost core depth profiles with measured $^{10}$Be and $^{26}$Al concentrations (black data points) with 1σ uncertainties (right). For all samples between 0.02–0.65 m depth, we used the average concentration of all five $^{10}$Be and $^{26}$Al measurements to represent the effect of cryoturbation of sediments in the active- and paleoactive-layer (see text). Note the rise in $^{10}$Be and $^{26}$Al concentrations below 2.09 m.

### 3.1.2 Lower Wright Valley borehole core

The WV14-I core is located in eastern Wright Valley at 326 masl (77.4252°S, 162.6664°E), ~2 km west of Wright Lower Glacier (Fig. 3). The core was recovered to a depth of 2.91 m (Fig. 5). The active-layer (0–0.28 m) above the ice-cemented permafrost consists of a thin armoured surface layer of desert

pavement (~0.02 m thick), and a layer of loose sand and pebbles (~0.26 m thick). Below 0.28 m depth, the recovered sediments comprised ice-cemented permafrost. The [10]Be and [26]Al depth profiles start on the armoured desert pavement. Two samples were collected from the active-layer and 10 from the ice-cemented permafrost (Fig. 5). The permafrost sediments are structureless, to thinly laminated, fine to coarse, and pebbly granular sands, which we interpret to be fluvial and aeolian deposits. Sediments within the core are derived from weathered granite, metasedimentary, dolerite and basalt origins. From 0–0.98 m depth, core sections were broken and loose sediment was recovered. Sediments recovered from 0.98–2.91 m were ice-cemented, except when encountering ice lenses. Several small ice lenses were recovered between 1.80–2.03 m depth. Hall et al. (2001) suggested sediments at lower Wright Valley are delta, shoreline and glaciolacustrine deposits associated with a large proglacial lake at the LGM and in the early Holocene (25–7 ka). Only four of the 10 permafrost core samples were successful in providing paired [10]Be and [26]Al concentrations (Fig 5; Table 1).

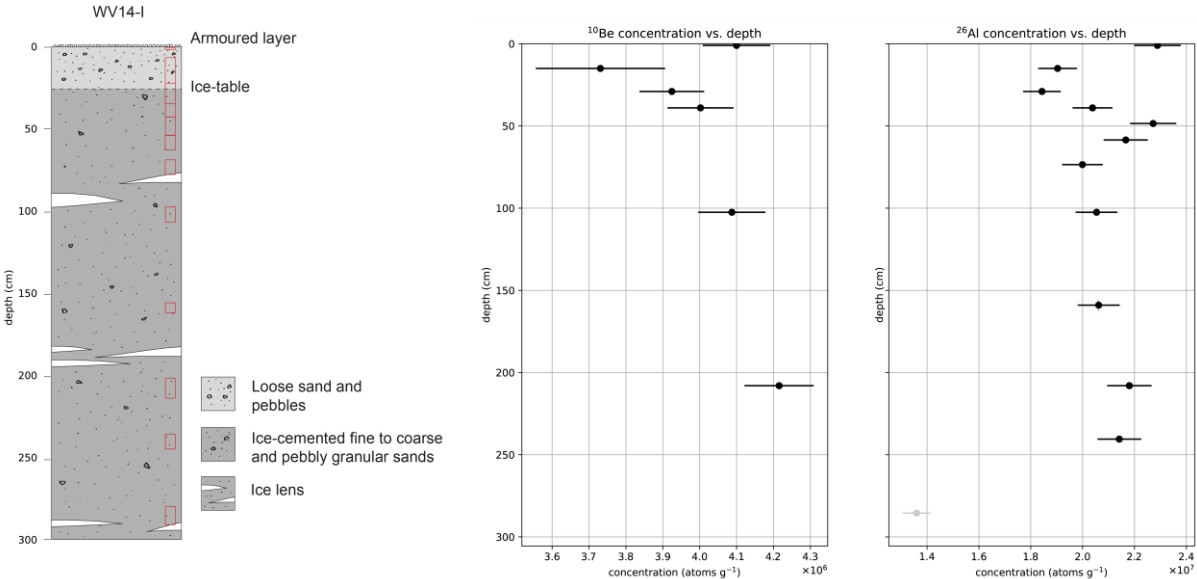

**Figure 5.** Lower Wright Valley (WV14-I) permafrost core sedimentology (left). Locations of cosmogenic nuclide samples shown in red boxes. The modern active-layer is from 0–0.28 m depth. lower Wright Valley (WV14-I) permafrost core depth profiles with measured [10]Be and [26]Al concentrations (black data points) with 1σ uncertainties (right).

## 3.2 Surface cobbles at Pearse Valley

Three granite cobble samples were collected for surface exposure analysis from Pearse Valley (Table 2; Fig. 2). We targeted perched cobbles, resting on larger flat boulders to minimise the possibility of post-depositional disturbance and hence best reflect deposition from retreating glacier ice or from surface deflation through sublimation. Samples that showed minimal weathering or fracturing were selected. The three cobbles were perched on larger host boulders (>1 m diameter) which were elevated above the local surface permafrost valley deposits (Fig. 6). Two samples (PV14-CS3-P1 and PV14-

CS3-P2) are small cobbles perched on the same host boulder, while the third sample (PV14-CS4-P1) is
a slightly larger cobble perched on a different host boulder less than 80 metres away.

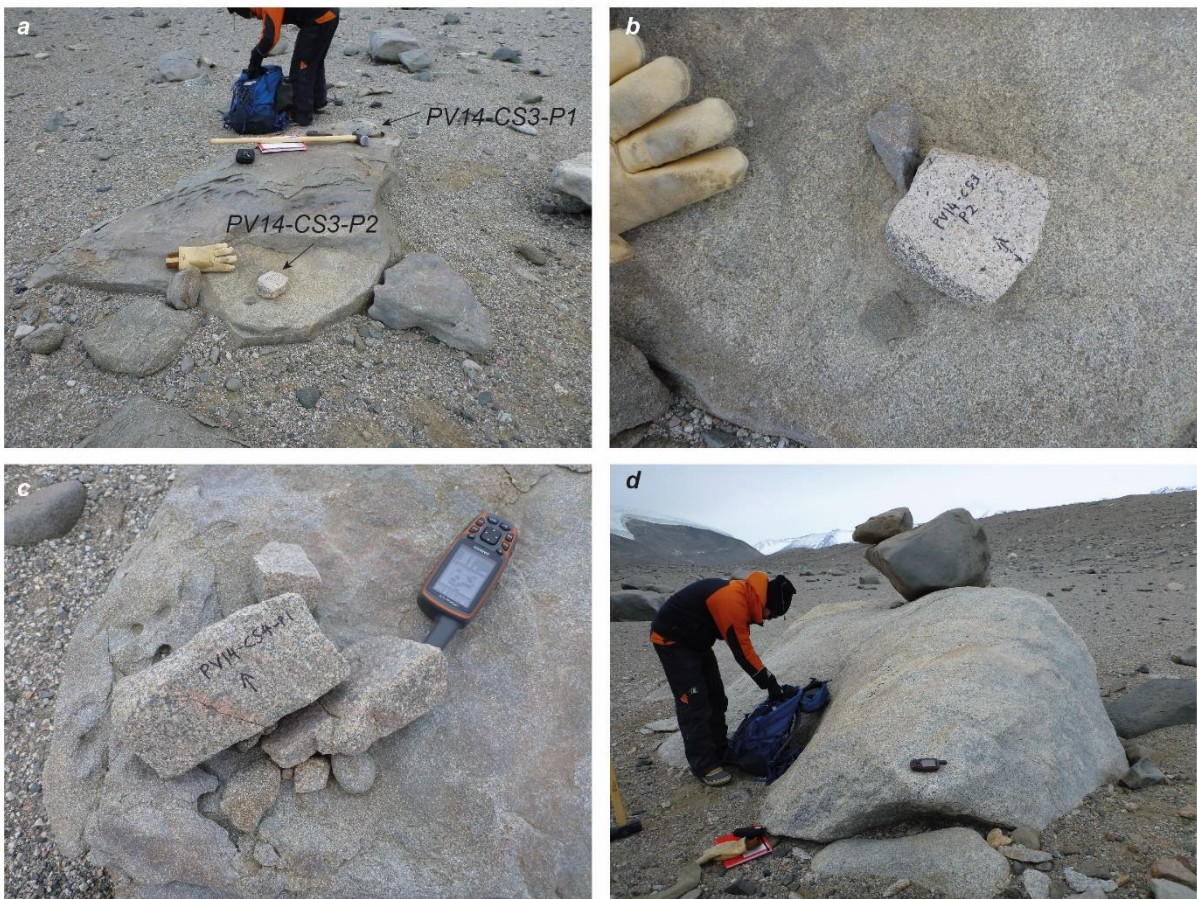


**Figure 6.** Boulders and cobbles from Taylor 2 Drift on the central northern side of Pearse Valley. (a)
PV14-CS3-P1 and PV14-CS3-P2 cobbles perched on a dolerite boulder. (b) Close view of PV14-CS3-
P2. (c) PV14-CS4-P1 cobble hosted on dolerite boulder. (d) A granite boulder, hosting a dolerite
boulder.

**3.3 Analytical methods**
Each core sample processed for cosmogenic nuclide analysis was heated at 100°C overnight to remove
ice and dry the sediment. Dried core samples, and cobble surface samples were crushed and sieved to
obtain the 250 – 500 µm fraction. Quartz was separated and purified using the hot phosphoric acid
method (Mifsud et al., 2013) and beryllium and aluminium were extracted from quartz via conventional
HF dissolution and ion exchange chromatography (Child et al., 2000). Isotope ratios were measured by
Accelerator Mass Spectrometry on the SIRIUS accelerator at the Australian Nuclear Science and
Technology Organisation (Wilcken et al., 2019).
Measured $^{10}Be/^{9}Be$ ratios were normalised to the 07KNSTD (KN-5.2) standard of Nishiizumi et al.
(2007) with a nominal $^{10}Be/^{9}Be$ ratio of 8560 x $10^{-15}$. Measured $^{26}Al/^{27}Al$ ratios were normalised to the
KNSTD (KN-4.2) standard of Nishiizumi (2004) with a nominal $^{26}Al/^{27}Al$ ratio of 30960 x $10^{-15}$. The
nuclide concentration data for the Pearse Valley and lower Wright Valley depth profiles, and perched
cobbles from Pearse Valley are shown in Tables 1 and 2, respectively. Full procedural $^{10}Be/^{9}Be$ blanks
were obtained using carrier solutions derived from dissolved beryl with known $^{9}Be$ concentrations
(1068 and 1048 μg/g (solution)) and resulted in ratios of $1.9 \pm 0.4$ x$10^{-15}$ and $1.3 \pm 0.3$ x$10^{-14}$. Blank
corrections to measured $^{10}Be/^{9}Be$ ratios amounted to <2%. Procedural $^{26}Al/^{27}Al$ blanks were processed
from standard reference ICP aluminium solutions (1000 μg/ml ±1%) and resulted in ratios $3.6 \pm 1.7$
x$10^{-14}$ and $1.3 \pm 0.6$ x$10^{-15}$. Blank corrections to measured $^{26}Al/^{27}Al$ ratios amounted to 4% to 35% for
Pearse Valley erratics and <1% for all other samples. Final errors in $^{10}Be$ and $^{26}Al$ concentrations are
obtained by quadrature addition of the final AMS analytical error (the larger of the total statistical or
standard mean error), a reproducibility error based on the standard deviation of the set of standard
reference samples measured during the run (typically 1-2% for either $^{10}Be$ or $^{26}Al$), a 1% error in Be
spike concentration and a representative 3% error for ICP Al concentration of the native $^{27}Al$ in the final
purified quartz powder. Unless otherwise stated, all analytical uncertainties are 1σ.
Surface exposure ages for the cobble samples were calculated using version 3 of the CRONUS-Earth
calculator (http://hess.ess.washington.edu/; Balco et al., 2008) using the LSDn scaling scheme (Lifton et
al., 2014) and the primary default calibration data set of Borchers et al. (2016) (Table 1). Complete
analytical data for all measurements are shown in Table S1, and data from surface samples are archived
on the ICE-D Antarctica database (http://antarctica.ice-d.org).

**3.4 Dual nuclide depth profile models and parameters**

$^{10}Be$ and $^{26}Al$ data from core samples at Pearse and lower Wright valleys were modelled as simple
exposure depth profiles (sensu Anderson et al., 1996). From a process perspective this assumes that (1)
the modelled sediment package is vertically well-mixed at the time of deposition such that inherited
nuclide concentration is constant with depth; (2) post-depositional sediment mixing is absent and
changes in bulk density do not occur over time; and (3) surface erosion is steady-state. While the
sedimentology of the cores clearly indicates that these assumptions were not fully realised, this simplified
model provides a useful tool for exploring the impact of various soil and permafrost processes while
providing useful chronologic constraints. We implemented a modified version of the Monte Carlo-based
code of Hidy et al. (2010) that allows profiles of both $^{10}Be$ and $^{26}Al$ to be modelled jointly (after Hidy et
al. (2018)). For shallow profiles in sediments, where non-unique solutions for exposure age and erosion
rate are likely, this approach allows estimation of exposure age and pre-depositional nuclide
concentration (i.e., inheritance) given reasonable observation-based constraint on erosion rate or net
erosion (e.g., Bergelin et al., 2022; Hidy et al., 2010, 2018; Mercader et al., 2012; Morgan et al., 2010).
The inheritance determined by the best-fit depth profile asymptote can be subtracted from the measured
values for each sample (Hidy et al., 2018). As described in Sect. 3.1 above, the upper ~0.3 m of both
cores consists of loose sandy sediment that is mobile or active. Fig. 7 shows a schematic evolution of a
cosmogenic nuclide depth profile over time with the added feature of a near-constant $^{10}$Be concentration
in a cryoturbated active-layer above ice-cemented permafrost. The presence of a surface mixed-layer
does not negate the assumption that these sediments were comprised of a combination of well mixed,
thick glacial tills, fluvial, and aeolian sediments that were deposited at a given time when the glaciers
retreated from each valley. However, consideration needs to be given on how to represent the measured
$^{10}$Be and $^{26}$Al concentrations in the surface mixed-layer with the depth profiles and resultant sensitivity
of the model outputs. We discuss these aspects in Sect. 4 below.
To ensure consistency with the cobble exposure ages, we obtain production rates applied in the depth
profile model from the CRONUS-Earth calculator. For the PV14-A core, we use a site-specific spallation
$^{10}$Be surface production rate of 8.40 atoms g$^{-1}$ (quartz) yr$^{-1}$, and a $^{26}$Al surface production rate of 59.7
atoms g$^{-1}$ (quartz) yr$^{-1}$. For the WV14-I core, we use a site-specific spallation $^{10}$Be surface production
rate of 7.47 atoms g$^{-1}$ (quartz) yr$^{-1}$, and a $^{26}$Al surface production rate of 53.2 atoms g$^{-1}$ (quartz) yr$^{-1}$.
These production rates were calculated using LSDn scaling (Lifton et al., 2014) and the primary
calibration data set of Borchers et al. (2016). These production rates yield $^{26}$Al/$^{10}$Be surface production
rate ratios of 7.11 and 7.12 for Pearse Valley and lower Wright Valley, respectively. We assume a
neutron attenuation length of $140 \pm 5$ g cm$^{-2}$, as used in previous Antarctic studies for $^{10}$Be and $^{26}$Al
(Bergelin et al., 2022; Borchers et al., 2016). Spallogenic production rate uncertainty has not been
included in the modelling. Muogenic production with depth, including an assumed 8% uncertainty,
followed Model 1A from Balco (2017). We assume bulk density to be constant with depth but sampled
from a normal distribution of $1.7 \pm 0.1$ g cm$^{-3}$ based on bulk density measured from two core samples
for loose sediment, and ice cemented permafrost. In most cases, the ice lenses were less than 5 cm thick.
The change of density in these thin ice lenses is not included in our assumed bulk density and we
acknowledge the small difference this assumption could have on the overall model outputs. Erosion rate
and net erosion were constrained between 0–0.4 cm/ka and 400 cm, respectively, based on field
observations described in Sect. 4.3. Within these constraints, exposure age, surface erosion rate, and
inheritance for $^{10}$Be and $^{26}$Al were simulated with uniform distributions, and model output was based on
n=100,000 acceptable depth profile solutions.






**Table 1.** Depth profile data from Pearse Valley and lower Wright Valley

| Sample name | Sample depth (m) | $^{10}$Be conc. ($10^6$ atoms g$^{-1}$)[a] | $^{26}$Al conc. ($10^6$ atoms g$^{-1}$)[b] | $^{26}$Al/$^{10}$Be ratio |
|---|---|---|---|---|
| **Pearse Valley** | | | | |
| PV14-SS-5 | 0.02 - 0.07 | 4.24 ± 0.095 | - | - |
| PV14-A-01 | 0.07 - 0.27 | 4.37 ± 0.097 | 18.67 ± 0.73 | 4.27 ± 0.19 |
| PV14-A-02 | 0.27 - 0.37 | 4.35 ± 0.097 | 17.97 ± 0.71 | 4.13 ± 0.19 |
| PV14-A-03 | 0.37 - 0.47 | 4.42 ± 0.098 | 19.63 ± 0.82 | 4.44 ± 0.21 |
| PV14-A-04 | 0.47 - 0.56 | - | 19.94 ± 0.78 | - |
| PV14-A-05 | 0.56 - 0.65 | 4.40 ± 0.098 | 18.28 ± 0.69 | 4.16 ± 0.18 |
| PV14-A-07 | 0.73 - 0.86 | 3.96 ± 0.089 | 17.95 ± 0.70 | 4.53 ± 0.20 |
| PV14-A-10 | 1.09 - 1.21 | - | 16.38 ± 0.64 | - |
| PV14-A-15 | 1.56 - 1.65 | 3.80 ± 0.085 | 15.09 ± 0.59 | 3.97 ± 0.18 |
| PV14-A-20 | 2.09 - 2.18 | 3.98 ± 0.080 | 17.50 ± 0.66 | 4.40 ± 0.19 |
| PV14-A-25 | 2.55 - 2.64 | 3.85 ± 0.086 | 16.70 ± 0.66 | 4.33 ± 0.20 |
| PV14-A-30 | 3.06 - 3.16 | - | 16.76 ± 0.66 | - |
| **Lower Wright Valley** | | | | |
| WV14-SS-01 | 0 - 0.02 | 4.10 ± 0.092 | 22.89 ± 0.89 | 5.58 ± 0.25 |
| WV14-I-01 | 0.07 - 0.23 | 3.73 ± 0.175 | 19.04 ± 0.75 | 5.10 ± 0.31 |
| WV14-I-02 | 0.23 - 0.35 | 3.92 ± 0.088 | 18.43 ± 0.72 | 4.70 ± 0.21 |
| WV14-I-03 | 0.35 - 0.43 | 4.00 ± 0.089 | 20.38 ± 0.77 | 5.09 ± 0.22 |
| WV14-I-04 | 0.43 - 0.54 | - | 22.72 ± 0.89 | - |
| WV14-I-05 | 0.54 - 0.63 | - | 21.66 ± 0.85 | - |
| WV14-I-07 | 0.69 - 0.78 | - | 19.99 ± 0.79 | - |
| WV14-I-10 | 0.98 - 1.07 | 4.09 ± 0.091 | 20.54 ± 0.81 | 5.02 ± 0.23 |
| WV14-I-14 | 1.56 - 1.62 | - | 20.62 ± 0.81 | - |
| WV14-I-20 | 2.02 - 2.14 | 4.22 ± 0.094 | 21.80 ± 0.86 | 5.17 ± 0.23 |
| WV14-I-23 | 2.36 - 2.45 | - | 21.41 ± 0.84 | - |
| WV14-I-29 | 2.80 - 2.91 | - | 13.60 ± 0.53 | - |


We assume a constant bulk density of $1.7 ± 0.1$ g cm$^{-3}$ based on bulk density measurements made on two core samples.

Topographic shielding is 0.9932 for Pearse Valley, and 0.9968 for lower Wright Valley, respectively.

[a] Normalised to the 07KNSTD (KN-5.2) standard of Nishiizumi et al. (2007).

[b] Normalised to the KNSTD (KN-4.2) standard of Nishiizumi (2004).


**Table 2.** Cosmogenic [10]Be and [26]Al concentrations and apparent exposure ages from Pearse Valley

| Sample name | Latitude (DD) | Longitude (DD) | Elevation (masl) | Sample thickness (cm) | Topographic shielding | [10]Be conc. (10^5 atoms g^{-1})[a] | [26]Al conc. (10^5 atoms g^{-1})[b] | Apparent [10]Be exposure age (ka)[c,d] | Apparent [26]Al exposure age (ka)[c,d] | [26]Al/[10]Be ratio | Erosion-corrected [10]Be exposure age (ka)[e] |
|---|---|---|---|---|---|---|---|---|---|---|---|
| PV14-CS3-P1 | -77.70737 | 161.55283 | 451 | 6 | 0.993 | $12.40 \pm 0.39$ | $76.57 \pm 4.48$ | $158 \pm 11$ (5) | $142 \pm 16$ (9) | $6.18 \pm 0.41$ | $174 \pm 13$ (6) |
| PV14-CS3-P2 | -77.70737 | 161.55283 | 451 | 3 | 0.993 | $5.36 \pm 0.15$ | $37.99 \pm 1.54$ | $65 \pm 4$ (2) | $66 \pm 7$ (3) | $7.09 \pm 0.35$ | $68 \pm 5$ (2) |
| PV14-CS4-P1 | -77.70747 | 161.55582 | 451 | 5 | 0.993 | $5.94 \pm 0.16$ | $33.71 \pm 5.14$ | $74 \pm 5$ (2) | $60 \pm 11$ (9) | $5.68 \pm 0.88$ | $77 \pm 5$ (2) |

All samples are granite cobbles and have a density of 2.65 g cm^{-3}.

[a] Normalised to the 07KNSTD (KN-5.2) standard of Nishiizumi et al. (2007).

[b] Normalised to the KNSTD (KN-4.2) standard of Nishiizumi (2004).

[c] Exposure ages calculated using the CRONUS-Earth calculator (http://hess.ess.washington.edu/math/), using the LSDn scaling scheme.

[d] Both internal and external uncertainties (shown at the $1\sigma$ level). Internal uncertainties (given in parentheses) are analytical uncertainties only and external uncertainties are absolute uncertainties and include production rate and scaling errors.

[e] Calculated using an erosion rate of 0.65 mm/ka.


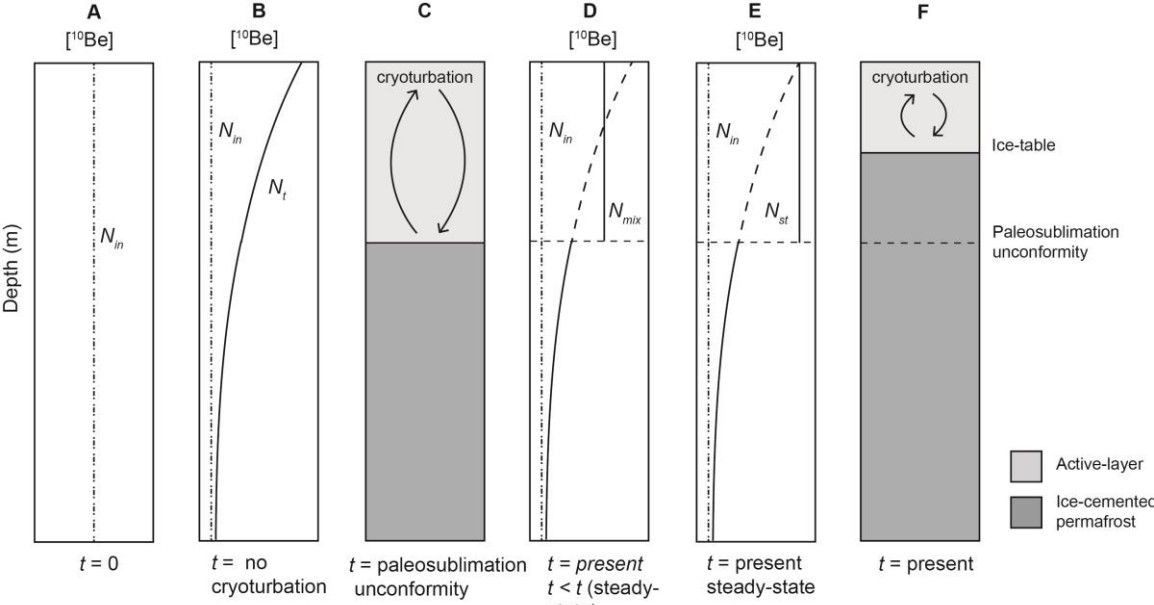


**Figure 7.** Schematic representation of a [10]Be depth profile in permafrost modified by active-layer cryoturbation. (a) Initial [10]Be profile (constant with depth) in well-mixed glacial till or sediment. All quartz grains are assumed to have been deposited with a common nuclide inheritance ($N_{in}$). (b) After prolonged exposure and in the absence of sediment mixing, an exponentially decreasing nuclide depth profile is obtained. (c) Permafrost profile during an interval where air temperature is warmer than present allowing near surface sediments to form an active-layer above the paleo-sublimation depth. Sediments below the unconformity are perennially frozen. (d & e) Vertical mixing via active-layer cryoturbation results in an average [10]Be value ($N_{mix}$) (d), and ($N_{st}$) with steady-state erosion (e). An exponentially decreasing [10]Be profile remains below the unconformity. (f) Present-day permafrost profile with shallower active-layer and ice-table than shown in (c).

## 4 Results

### 4.1 Cosmogenic nuclide depth profiles

Both the Pearse Valley (Fig. 4), and lower Wright Valley (Fig. 5) depth profiles share two common observations. Neither depth profile displays a marked exponential decrease in measured nuclide concentration over the full ~3 m core depth profile, and both cores have shallow, active mixed-layers where measured nuclide concentrations are effectively constant.

In the Pearse Valley permafrost core, there is a marked decrease in all [10]Be and [26]Al concentrations for samples below ~0.65 m depth. However, the reduction in [10]Be (and [26]Al) between shallow (active-layer) and deep samples from only ~4.4 to ~3.8 x10[6] atoms g[-1] (and respectively from ~19.9 to ~15.1 x10[6] atoms g[-1] for [26]Al) indicates a high inherited cosmogenic concentration supporting a marginal post-depositional increase of [10]Be and [26]Al. Moreover, the average [26]Al/[10]Be ratio which ranges between 4.0 to 4.5 suggests a long history of total exposure and burial for these permafrost sediments (i.e., in addition

to their presence in the core as permafrost). One feature worthy of note, is the distinct increase in both $^{10}$Be and $^{26}$Al for the deepest three samples below 2.09 m depth compared to samples <1.65 m depth, suggesting that the Pearse Valley permafrost core may not have been a single depositional event. In contrast, the lower Wright Valley depth profiles for $^{10}$Be and $^{26}$Al show more scatter than the Pearse Valley depth profiles and there is no decrease in concentration with depth. Effectively the lower Wright Valley profile is depth independent with a $^{10}$Be concentration at ~4.0 x10$^6$ and a $^{26}$Al concentration at ~20.3 x10$^6$ atoms g$^{-1}$. The magnitudes of the concentrations for Pearse and Wright valleys are remarkably similar, as is the range in $^{26}$Al/$^{10}$Be ratio from 4.7 to 5.6, suggesting that lower Wright Valley permafrost sediments have had a similar total exposure-burial history as Pearse Valley sediments.

These depth profiles present complications to any modelling aiming for non-unique solutions of deposition age and surface erosion due to the presence of a surface mixed-layer and marginal (in Pearse Valley) to near absent (in lower Wright Valley) post-depositional build-up of $^{10}$Be and $^{26}$Al in the shallow subsurface sediments. We note that applying a depth profile model that assumes nuclide concentration attenuation to a profile that contains a surface mixed-layer and depth concentration inversions has limitations with respect to chronological information. In the following sections we describe the modified depth modelling exercises taken to accommodate the complication presented in the Pearse Valley and lower Wright Valley data sets.

## 4.2 Minimum age estimate for Pearse Valley core

Prior to any depth profile modelling, a simple calculation was carried out to estimate the depositional age of the upper ~0.65 m of the Pearse Valley permafrost by comparing maximum and minimum nuclide concentrations. Assuming zero erosion and a surface production rate determined at the coring site, a minimum 'exposure age' ($t_{min}$) can be calculated using the following equation:

$$t_{min} = (N_{max} - N_{min})/ P \qquad (1)$$

Where $N_{max}$ is the absolute maximum $^{10}$Be concentration, $N_{min}$ is the absolute minimum $^{10}$Be concentration (assumed inheritance) for all mixed sediments, and $P$ is the production rate (atoms g$^{-1}$) at the sample site. The absolute maximum and minimum $^{10}$Be concentrations for the Pearse Valley depth profile using equation 1 are reported in Table 3. Equation 1 yielded a minimum deposition age of ~74 ka for the Pearse Valley core (Table 3).

**Table 3.** Maximum and minimum $^{10}$Be concentrations and minimum deposition age for the Pearse Valley core.

| Borehole | $P$ | $N_{max}$ ($10^6$ atoms g$^{-1}$) | $N_{min}$ ($10^6$ atoms g$^{-1}$) | Min age (ka) |
|----------|-----|-----------------------------------|-----------------------------------|--------------|
| PV14-A | 8.4 | 4.42 | 3.80 | 74 |

### 4.3 Cosmogenic nuclide depth profiles at Pearse Valley

Below the surface mixed-layer, between 0.65 m and 1.65 m, both $^{10}$Be and $^{26}$Al concentrations display attenuation with depth. Below 1.65 m, the attenuation is interrupted by a considerable increase in nuclide concentrations from 2.09 m depth. This suggests that the depth profile is of a composite structure, which is supported by the observation that ice lenses appearing at ~0.7 m, and at ~1.70–1.80 m (see Fig. 4), are associated with distinct changes in $^{10}$Be and $^{26}$Al concentrations. No acceptable depth profile model fit was obtained when all measured $^{10}$Be and $^{26}$Al concentrations were included as a single depositional episode (see Fig. S1). Hence, consideration was given to restrict our depth profile model to only fit samples from 0.02 to 1.65 m depth, and how to incorporate the surface mixed-layer with the depth profile.

The five $^{10}$Be and five $^{26}$Al nuclide concentrations from 0.02–0.65 m exhibit a uniform concentration with depth with averages of $4.36 \pm 0.10$ x10$^6$ atoms g$^{-1}$ and $1.89 \pm 0.07$ x10$^7$ atoms g$^{-1}$, respectively, with no attenuation, indicating that these upper sediments have been vertically mixed (or possibly deposited sufficiently recently so that nuclide depth profiles effectively reflect only inheritance without significant post-depositional production). In continuously vertically mixed surface soils (such as those in the McMurdo Dry Valleys), where mixing times are short compared to radionuclide decay rates, the average production rate in the mixed-layer is constant with depth (Granger and Riebe, 2014). Under these conditions, the average cosmogenic nuclide concentration in the mixed-layer will attain a constant value at erosional equilibrium (Fig. 7). Hence, we use the mean $^{10}$Be and $^{26}$Al concentrations in the upper 0.65 m to approximate the surface mixing processes that resulted in the uniform profile. Fig. 8 shows the model best-fit to samples from 0.02–1.65 m, with all samples between 0.02 and 0.65 m depth converging to a single mean concentration in order to determine the younger depositional phase. When solving for the four free parameters, namely, age, erosion rate, $^{10}$Be and $^{26}$Al inheritance, the best-fit modelled nuclide concentrations for the PV14-A depth profile when restricted to samples from 0.02 to 1.65 m depth, falls within the 25$^{th}$ to 75$^{th}$ percentile of the measured concentrations. The reduced chi-squared statistical test for the best-fit to a profile using a mean concentration for the surface mixed-layer with the upper sediment samples (0.02 to 1.65 m depth) gives a value of 0.88 with three degrees of freedom (n=7) which is significantly better than the reduced chi-squared value of 2.70 with 16 degrees of freedom (n=20) for the full profile using all nuclide measurements (0.02 – 3.16 m) (see SD3), confirming our modified approach improved model fitting. We constrained the erosion rate of the depth profiles using information from surface cobble PV14-CS3-P2 which sits ~10–20 cm above the desert pavement and has a minimum exposure age of 65 ka (Fig. 6a). Based on this observation we can assume a maximum surface lowering rate of ~0.3 cm ka$^{-1}$. Using this field observation, we applied a conservatively high erosion rate limit of 0.4 cm ka$^{-1}$ for our depth profile modelling. The solutions yield

most probable $^{10}$Be and $^{26}$Al inheritance concentrations of 3.59 x 10$^6$ and 1.42 x 10$^7$ atoms g$^{-1}$,
respectively (Fig. 8; Fig S2) and constrain the depositional age of the sediment (<1.65 m depth) at 180
$^{+20}$/ $_{-40}$ ka (Fig. 9), and an erosion rate of 0.24 $^{+0.10}$ / $_{-0.09}$ cm ka$^{-1}$ (Fig. S2). By inference, the lower part
of the profile (>2.09 m depth) predates the sediments above and must be deposited before ~180 ka.

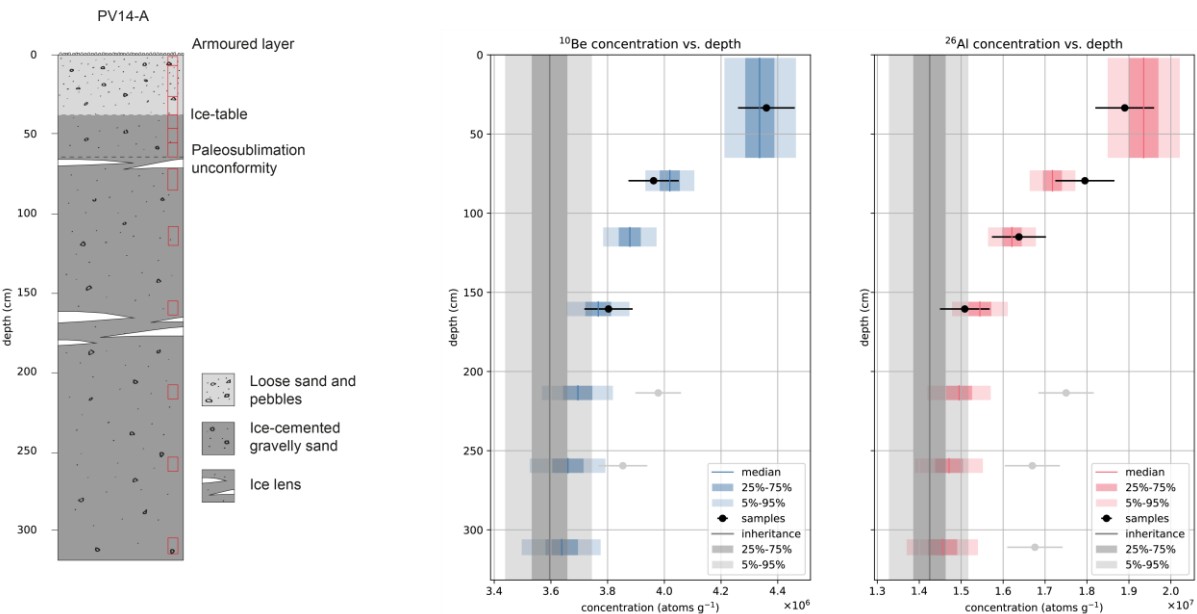


**Figure 8.** Pearse Valley (PV14-A) permafrost core sedimentology (left). Locations of cosmogenic
nuclide samples shown in red boxes. Pearse Valley (PV14-A) permafrost core depth profiles with
measured $^{10}$Be and $^{26}$Al concentrations (black data points) with 1σ uncertainties (right). For all samples
between 0.02–0.65 m depth, we used the average concentration of all five $^{10}$Be and $^{26}$Al measurements
to represent the effect of cryoturbation of sediments in the active-layer. Blue ($^{10}$Be) and red ($^{26}$Al) boxes
show simulated nuclide concentrations at each depth. $^{10}$Be and $^{26}$Al concentrations (grey data points)
below 2.09 m were not included in the model.

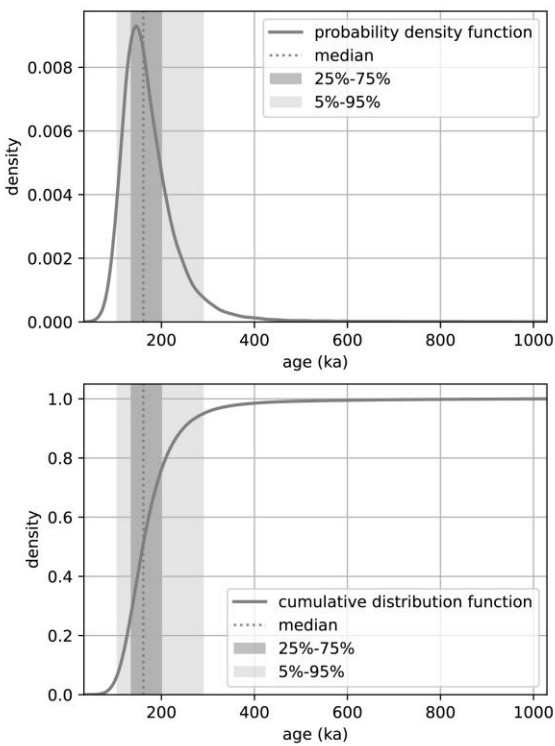


**Figure 9.** Probability density function, and cumulative distribution function for exposure age, using dual-nuclide depth profile modelling between 0.02 – 1.65 m depth for PV14-A.


### 4.4 Cosmogenic nuclide depth profiles at lower Wright Valley

The $^{10}$Be and $^{26}$Al depth profiles from the permafrost core and overlying active-layer used for depth profile modelling at lower Wright Valley is shown in Fig 10. The lower Wright Valley $^{10}$Be and $^{26}$Al concentration profiles exhibit near-constant concentrations with depth, with average values of 4.01 ± 0.10 x10$^6$ atoms g$^{-1}$ and 2.08 ± 0.08 x10$^7$ atoms g$^{-1}$, respectively. The absence of a discernible exponential attenuation indicates all sediments in the depth profile are either continuously vertically mixed after deposition, or are sufficiently young so that post-depositional nuclide production is negligible relative to inheritance.

The depth profile model does not work well for non-attenuating profiles and usually fails to give well-constrained results. The modelled nuclide concentration depth profiles do not fit within the 5$^{th}$ to 95$^{th}$ percentile for our measured concentrations in the lower Wright Valley depth profile (Fig. 10). The solutions yield most probable $^{10}$Be and $^{26}$Al inheritance concentrations of 4.03 x 10$^6$ and 2.06 x 10$^7$ atoms g$^{-1}$, respectively (Fig. 10; Fig. S4). Our simulations yield the depositional age of the permafrost at 4.4 $^{+8.2}$/$_{-4.2}$ ka (5$^{th}$ to 95$^{th}$ percentile), and an erosion rate of 0.2 $^{+0.18}$/$_{-0.18}$ cm ka$^{-1}$ (Fig. S4).

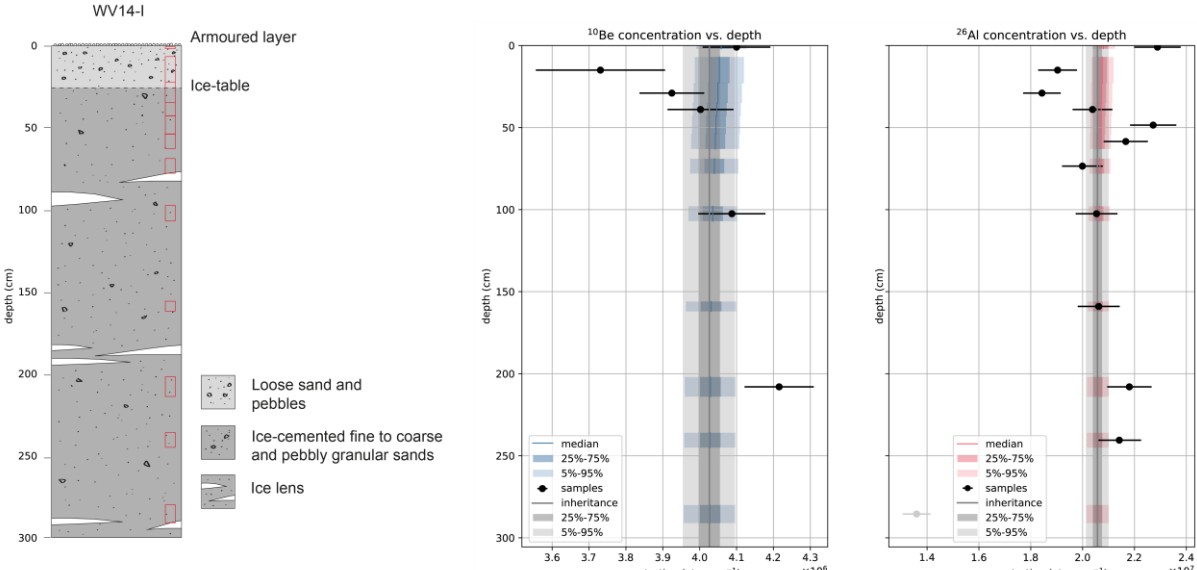

**Figure 10.** Lower Wright Valley (WV14-I) permafrost core sedimentology (left). Locations of cosmogenic nuclide samples shown in red boxes. Lower Wright Valley (WV14-I) permafrost core depth profiles with measured [10]Be and [26]Al concentrations (black data points) with 1σ uncertainties (right). Blue ([10]Be) and red ([26]Al) boxes show simulated nuclide concentrations at each depth.

## 4.5 Surface exposure ages and erosion rates at Pearse Valley

Boulders and cobbles of granite, gneiss, Beacon sandstone and dolerite pepper the Pearse Valley floor, forming a thin, patchy drift overlying an older, well-weathered relict drift surface. Some boulders lodged in the relict drift host smaller perched boulders, cobbles, and pebbles on their surfaces, indicating deposition of perched clasts occurred after the most recent retreat of Taylor Glacier (Fig. 6). Our surface exposure chronology is based on three granitic cobbles on the northern side of the central valley floor (Table 2, Fig. 2). Two samples (PV14-CS3-P2 and PV14-CS4-P1) yielded minimum zero-erosion [10]Be exposure ages of 65 ± 4 ka and 74 ± 5 ka (1σ external errors), respectively, whereas the third sample (PV14-CS3-P1) yielded an older age of 158 ± 11 ka, presumably affected by inheritance (Table 2). The three [26]Al/[10]Be concentration ratios range from 5.7 to 7.1 and when plotted on [10]Be-[26]Al/[10]Be diagram, are consistent with a simple constant exposure within their 1σ error ellipses (Fig. 11). One sample (PV14-CS4-P1) suggests a burial age ranging from 0 up to ~900 ka burial, the result of a large error in measured [26]Al concentration. Given inheritance is stochastic, we infer the two lowest consistent ages represent the minimum inheritance, and we take them to be our best estimate to represent zero-erosion exposure ages for the cobbles. While this assumption of zero erosion makes negligible difference for LGM and younger ages, we evaluate the influence of surface erosion on the exposure ages above using known erosion rates reported from Antarctica and geological evidence from the sites. Bedrock and regolith erosion rates in the McMurdo Dry Valleys range from 0.1–4 mm/ka (Putkonen et al., 2008; Summerfield et al., 1999). A compiled study across Antarctica showed that granite populations have a

mean erosion rate of 0.13 mm/ka, and in the Dry valleys, a max erosion rate of 0.65 mm/ka (Marrero et
al., 2018). Applying the max erosion rate (0.65 mm/ka) from granite surfaces in the McMurdo Dry
Valleys, erosion corrected [10]Be exposure ages of our granitic cobbles resulted in 174 ± 13 ka (PV14-
CS3-P1), 68 ± 5 ka (PV14-CS3-P2) and 77 ± 5 ka (PV14-CS4-P1) (1σ external errors; Table 2). The
cobble sample PV14-CS3-P2 displays minimal edge rounding which suggests negligible erosion and is
unlikely to be much older than the zero-erosion age.

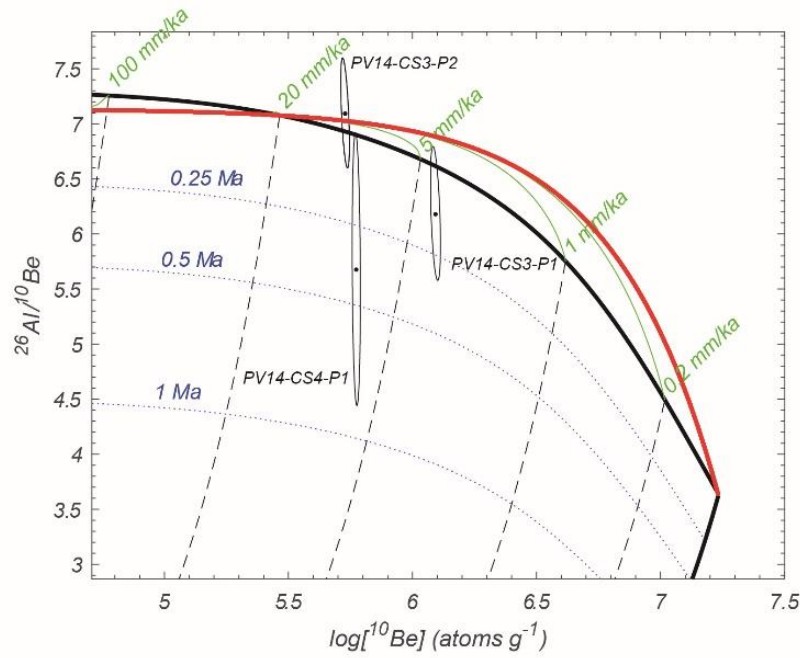

**Figure 11.** Two-isotope plot of Pearse Valley cobbles using the time-dependent LSDn scaling scheme
of Lifton et al. (2014) and the primary default calibration data set of Borchers et al. (2016). Measured
nuclide concentrations are shown with 1σ uncertainties. Burial isochrons (dotted lines), decay
trajectories (dashed), the exposure-erosion region (bounded by black and red lines), and steady-state
erosion loci (green) are shown.

**5 Discussion**
**5.1 Depositional and permafrost processes at Pearse Valley**
Depth profile modelling suggests that the permafrost sediments underlying Taylor 2 Drift, at Pearse
Valley, predate MIS 5. At the PV14-A permafrost core site, the present-day active-layer comprises a
desert pavement surface and layer of loose vertically mixed sediments to a depth of ~0.37 m, positioned
above ice-cemented permafrost sediments. The interface between this active-layer and the ice-cemented
permafrost represents a sublimation unconformity. [10]Be and [26]Al concentrations are constant throughout
the active layer and down to ~0.65 m depth in the permafrost. However, there is a discernible decrease
in [10]Be and [26]Al concentrations in the permafrost below ~0.65 m depth alongside an ice horizon (Fig. 4).
Such ice horizons are indicative of a paleosublimation unconformity, and suggests the sediments

experienced intervals that are warmer than present-day during or after deposition. This [10]Be reduction cannot be explained by active-layer cryoturbation, as the present-day active-layer is only 0.37 m deep. Lapalme et al. (2017) suggested that in the upper ~0.5 m of a soil profile, ice can accumulate and sublimate due to changing ground surface temperature and humidity conditions. Below ~0.5 m depth, ice will progressively increase over time. Therefore, a paleosublimation unconformity can be inferred by the increase in ice content from 0.6 to 0.4 m depth, which records the maximum predicted ice table depth (Lapalme et al., 2017). Therefore, we suggest the [10]Be reduction between the sediments above and below 0.65 m represent a paleosublimation unconformity which probably formed when the active-layer was thicker than present. However, we cannot rule out that the fluctuation of the present-day active-layer depth through summer months could represent annual variability of the active-layer. Although, the lack of active-layer thickness exceeding >50 cm depth in low elevation McMurdo Dry Valleys locations (Bockheim et al., 2007) suggests this is unlikely in Pearse Valley which is further inland and at higher elevation. Gravimetric water content is relatively high in near-surface permafrost in the McMurdo Dry Valleys (Lacelle et al., 2022), and water content in permafrost influences the susceptibility of cryoturbation. Our depth profile model indicates that the upper section of the Pease Valley permafrost sediments (<1.65 m) was likely deposited at $180 ^{+20}/_{-40}$ ka, which does not contradict the exposure ages of the thin, patchy drift (~65–74 ka). Our measured nuclide concentrations at >2.09 m depth largely differ from the upper section and do not fit the simulated depth profile constrained between 0.02 and 1.65 m depth (Fig. 8). The increase in nuclide concentrations at >2.09 m depth relative to the samples between 1.09–1.65 m depth, alongside the presence of several small ice lenses between 1.57–1.87 m depth, suggest these sediments were deposited during an earlier depositional event before ~180 ka. If the lower set of ice lenses (1.57–1.87 m depth) represent the bottom of a paleoactive-layer, this would imply ~0.5–0.8 m of erosion prior to the most recent episode of sediment deposition above 1.65 m. The sedimentology of the core lacks evidence to suggest if this scenario is plausible or not. The attenuating depth profile (>0.65 m depth) shows that sediments at Pearse Valley have not been vertically mixed since MIS 5, but surface mixing has occurred to at least 0.65 m depth in the last ~74 ka.

There are several complications regarding modelling the permafrost depth profiles that limit the reliability in calculating deposition age and surface erosion rates. Firstly, Pearse Valley is episodically covered by ice from Taylor Glacier advances. During periods of ice cover, vertical mixing does not occur. Secondly, using a mean concentration for the measured samples in the surface mixed-layer (0.02–0.65 m depth) is equivalent to assuming the mean value can represent a constant well-mixed layer. We acknowledge using a mixing model (e.g. Knudsen et al., 2019; Lal & Chen, 2005) for the depth profile data would allow an alternate approach, and may provide an improved fit, among many possible scenarios. However, given the complexity of these data and uncertainty of ice cover by Taylor Glacier, which cannot be incorporated in other mixing models, simply using the mean concentration within the upper 0.65 m is a reasonable approximation.

**5.2 Exposure-burial history of sediments in Pearse Valley and lower Wright Valley**

While nuclide depth profiles indicate the most recent depositional history of the permafrost sediment, $^{26}$Al/$^{10}$Be ratio data provides an additional insight regarding the total history of the sediment. When $^{26}$Al/$^{10}$Be is plotted against $^{10}$Be concentration on a two-isotope diagram (Fig. 12), a minimum total exposure-burial period can be inferred on the assumption that the sample experienced only one cycle of continuous exposure followed by continuous deep burial. At the Pearse Valley site, the two-isotope plot indicates that all sediments, regardless of their depth, have $^{26}$Al/$^{10}$Be ratios ranging from 3.97 to 4.53, resulting in a minimum ~800 ka simple exposure (at zero erosion), and minimum ~400 ka burial, with a total exposure-burial history of at least 1.2 Ma. At the lower Wright Valley site, $^{26}$Al/$^{10}$Be ratios for all samples range from 4.70 to 5.58, resulting in a minimum ~900 ka simple exposure, and minimum ~300 ka burial, with a total exposure-burial history of at least 1.2 Ma. These exposure-burial histories from the two-isotope plots for the Pearse and lower Wright valleys depth profiles assume that the surface production rate at each of the core elevations represents a minimum value.

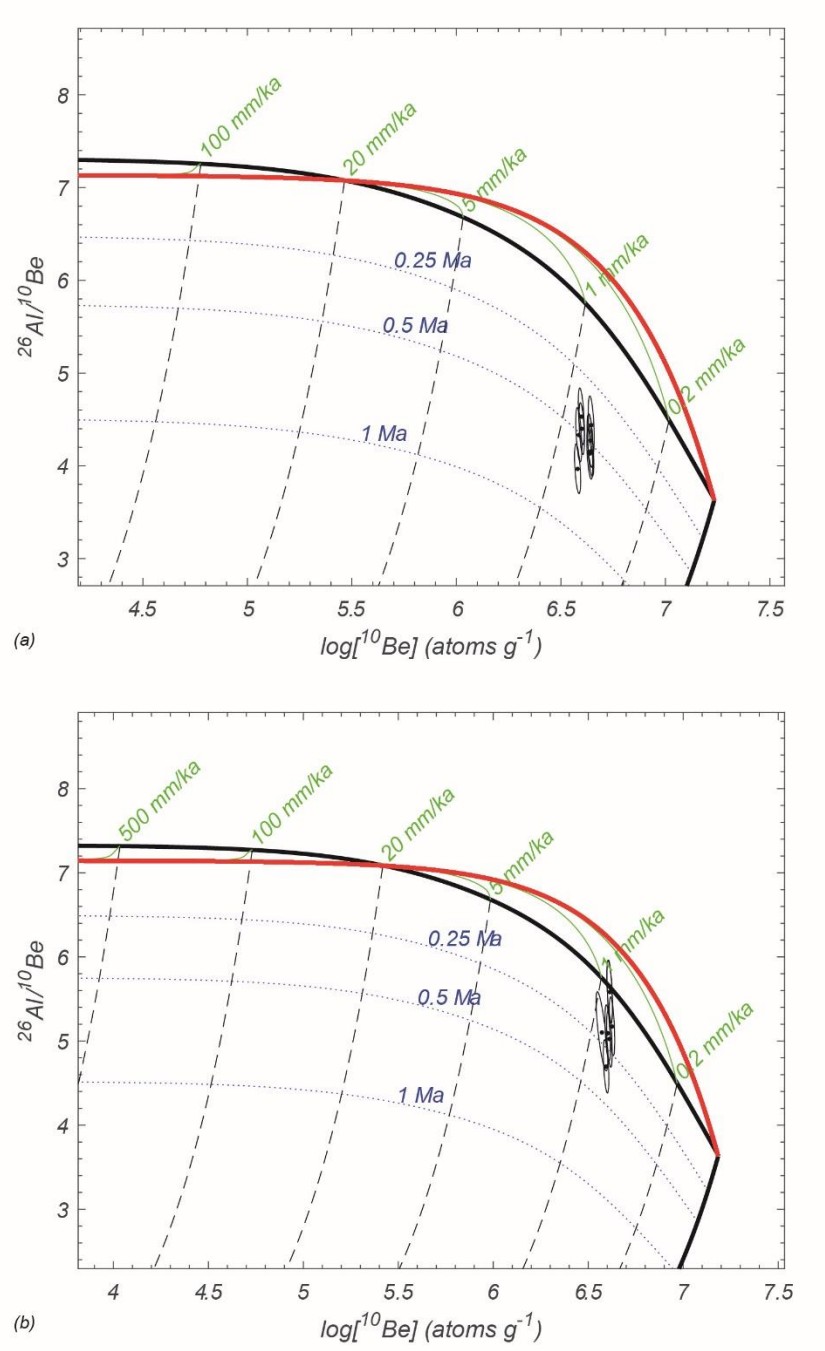

**Figure 12.** Two-isotope plot of Pearse Valley (a) and lower Wright Valley (b) depth profiles using the time-dependent LSDn scaling scheme of Lifton et al. (2014) and the primary default calibration data set of Borchers et al. (2016). Measured nuclide concentrations are shown with 1σ uncertainties. Burial isochrons (dotted lines), decay trajectories (dashed), the exposure-erosion region (bounded by black and red lines), and steady-state erosion loci (green) are shown. The exposure-erosion regions are produced using the surface production rates of 8.40 atoms g$^{-1}$ yr$^{-1}$ for Pearse Valley, and 7.47 atoms g$^{-1}$ yr$^{-1}$ for lower Wright Valley, respectively.

Depth profile modelling of near-surface sediments at both permafrost core sites represent the most recent phase of their depositional histories. Pearse Valley permafrost sediments were emplaced ~180 ka, using

a best-fit surface erosion rate of 0.24 cm ka$^{-1}$. For lower Wright Valley, where $^{10}$Be and $^{26}$Al
concentrations do not attenuate, depth profile modelling is not useful in determining age. Instead, we
estimate a maximum deposition age of <25 ka. This age represents the time required to change $^{10}$Be and
$^{26}$Al above the initial inheritance level for near-surface samples by 5% - a change outside AMS $^{10}$Be and
$^{26}$Al measurement error. However, our $^{26}$Al/$^{10}$Be ratios at both sites suggest that these sediments have
much longer total exposure-burial histories of at least 1.2 Ma, which most likely involves multiple
recycling episodes of exposure, deposition, burial, and deflation prior to deposition at their current
locations. Million-year exposure-burial recycling periods of sediments in the McMurdo Dry Valleys was
also observed in shallow (<1 m) pits from the Packard Dune fields in Victoria Valley (Fink et al., 2015).
In summary, Pearse Valley sediments are old, have a complex exposure-burial history >1.2 Ma, were
recently deposited ~180 ka, and their shallow-surface sediments (<0.65 m depth) were subject to active-
layer mixing. Lower Wright Valley sediments are equally old, with a similar exposure-burial history, but
were deposited and mixed after the LGM.

### 5.3 Fluctuations of Taylor Glacier in Pearse Valley during MIS 5


Thin, patchy drift at Pearse Valley is a discontinuous peppering of boulders and cobbles superimposed
on older loose sandy sediments, reworked clasts, and underlying permafrost sediments (Fig. 6).
Exposure ages of surface cobbles perched on large boulders confirm that this thin, patchy drift was
deposited by a retreating cold-based Taylor Glacier during MIS 5a, and the MIS 5 / 4 transition, on the
northern valley floor of central Pearse Valley, whereas the underlying permafrost sediments were
deposited at ~180 ka or earlier.
Our surface cobble geochronology is in agreement with the minimum U/Th ages for the extent of
proglacial Lake Bonney, which suggest retreat of Taylor Glacier following MIS 5c and 5a advance (Fig.
13; Higgins et al., 2000a), and the tentatively dated western section of the rock glacier derived from
$\delta^{18}$O in buried ice in northern Pearse Valley (Swanger et al., 2019). These data suggest Pearse Valley
was largely or partially glaciated throughout MIS 5c and 5a.
Retreat of the Taylor Glacier lobe in Pearse Valley possibly continued after 65 ka. Timing of retreat
after 65 ka, until the Last Glacial Maximum, where Taylor Glacier was at a minimum position, remains
unknown. Advance and retreat cycles during MIS 5, the final retreat of Taylor Glacier during MIS 5a,
and between the MIS 5 / 4 transition and the LGM for Taylor Glacier, could be better constrained by
exposure dating more drift deposits with larger spatial coverage from Pearse Valley.

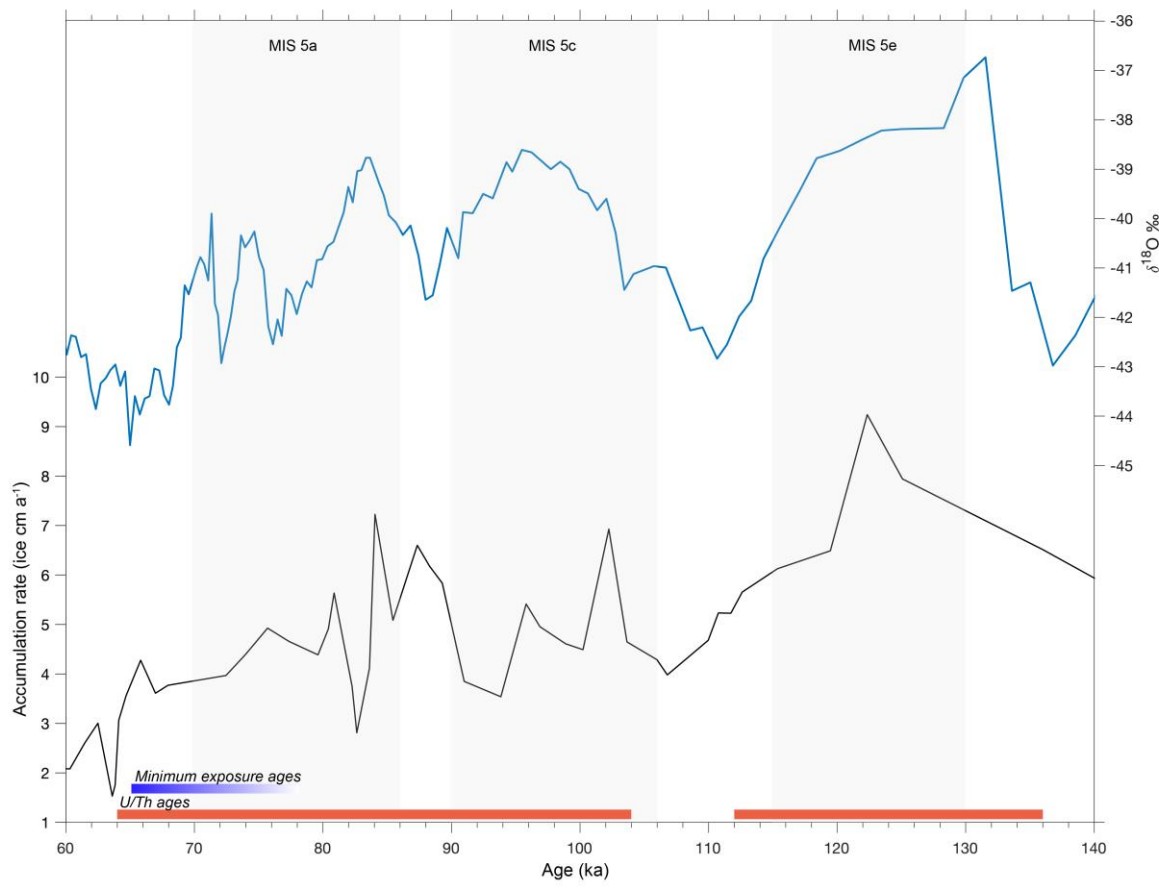

**Figure 13**. Snow accumulation rate (ice cm a$^{-1}$) determined from $^{10}$Be and $\delta^{18}$O record from Taylor Dome during MIS 5 (Steig et al., 2000). U/Th ages from algal carbonates (red bands, Higgins et al., 2000a) coincide with warm MIS substages 5e, 5c and 5a with increased accumulation rates at Taylor Dome. This is consistent with our minimum exposure ages (blue band) which show retreat of Taylor Glacier in Pearse Valley during MIS 5a, and the MIS 5 / 4 transition.

## 5.4 Advance and retreat of outlet and alpine glaciers during interglacial periods

Our new data has implications regarding the relationship between outlet and alpine glacier behaviour, regional paleoclimate, and the extent of sea ice and open ocean conditions in the Ross Sea. Snow accumulation rate, atmospheric temperature, and duration of precipitation appear to be the major controls governing the advance and retreat of Taylor Glacier during previous warm intervals (Fig. 13). In central Taylor Valley, substage 5a and 5c sediments bury 5e sediments suggesting Taylor Glacier responds to regional changes over millennial timescales (Higgins et al., 2000a). The Taylor Glacier advances in central Taylor Valley during substages 5e, 5c and 5a correspond with increased accumulation in Taylor Dome (Higgins et al., 2000a; Steig et al., 2000). Our exposure ages indicate the retreat of Taylor Glacier in Pearse Valley occurred at ~65–74 ka, during the MIS 5 / 4 transition and is consistent with the retreat in central Taylor Valley. The presence of a lobe of Taylor Glacier in Pearse Valley throughout MIS 5 is likely linked to prolonged interglacial climate conditions. The interglacial-mode climate, where austral westerlies are in a poleward-shifted position for prolonged periods during

MIS 5, is associated with periods where $CO_2$ concentrations were above ~230 ppm, the glacial-
interglacial $CO_2$ threshold proposed by Denton et al. (2021).
Yan et al. (2021) suggested that peak accumulation rates occurred at ~128 ka in Southern Victoria Land
and are associated with reduced sea ice and possibly retreat of the Ross Ice Shelf. The study suggested
by ~125 ka, the Ross Ice Shelf had returned to a configuration comparable to present day. However, a
reduction of sea ice may have enabled increased moisture delivery over Taylor Dome during MIS 5c
and 5a. As Higgins et al. (2000a) suggested, increased precipitation over Taylor Dome during MIS 5a
and 5c appears to have caused a subsequent readvance of Taylor Glacier. We acknowledge, this
hypothesis is speculative and requires further testing of temperature, and atmospheric circulation in
response to reduced sea ice extent and perhaps a reduction of the Ross Ice Shelf by climate models.

**6 Conclusions**
We applied cosmogenic nuclide analysis to ~3 m permafrost depth profiles in Pearse and lower Wright
valleys of the McMurdo Dry Valleys to determine their age of deposition, permafrost processes and
landscape evolution. Additionally, cosmogenic surface exposure dating of surface cobbles perched on
large boulders at Pearse Valley provide reliable ages for the Taylor 2 Drift. Paired $^{10}$Be and $^{26}$Al depth
profiles at Pearse Valley show a mixed-layer in the upper ~0.65 m of sediment since ~74 ka, and depth
profile modelling for near-surface permafrost deposits to 1.65 m depth reveals a deposition age of 180
$^{+20}/_{-40}$ ka that predates MIS 5. The sharp reduction in $^{10}$Be concentrations at ~0.65 m depth, and presence
of increased ice content reveals a paleosublimation unconformity, and suggests that these upper
sediments have undergone active-layer cryoturbation. The near-surface sediment (including the surface
mixed-layer 0.02–0.65 m and permafrost at 0.65–1.65 m depth) in central Pearse Valley has been
deposited at ~180 ka based on our depth profile model, whereas, at >2.09 m depth the depositional age
of the sediment must be earlier than ~180 ka. To compare processes of sediment evolution at Pearse
Valley with a lower elevation, and more coastal environment, we also applied $^{10}$Be and $^{26}$Al nuclide
analysis to permafrost depth profiles at lower Wright Valley. While the current deposition at the latter
site occurred more recently (<25 ka), total exposure-burial histories from the two sites consistently
show these sediment repositories have experienced multiple glacial-interglacial cycles achieved through
the recycling of sediments for at least 1.2 Ma. Our $^{10}$Be and $^{26}$Al derived surface exposure ages from
cobbles emplaced on large boulders embedded in the valley floor of Pearse Valley located ~3 km from
Taylor Glacier lobe give a minimum zero erosion age of ~65 to 74 ka for deposition of the thin, patchy
drift, indicating that Taylor Glacier retreated from Pearse Valley during MIS 5 / 4 transition. These data
support antiphase behaviour between outlet and alpine glaciers in the McMurdo Dry Valleys region and
ice extent in the Ross Sea, and suggest a causal mechanism where cold-based glacier advance and retreat
is controlled by moisture availability and drying, respectively due to ice retreat and expansion in the

Ross Sea. Our work is consistent with geochronology from central Taylor Valley, supporting advance and retreat cycles of Taylor Glacier during MIS substages 5c and 5a (Higgins et al., 2000a), corresponding with increased accumulation at Taylor Dome (Steig et al. 2000).

**Code availability**

The code used for depth profile modelling is available by request from the corresponding author.

**Data availability**

All data described in the paper are included in the Supplement.

**Author contributions**

JTHA, GSW, AA, and ND conducted the field work and sample collection. JTHA did the sample preparation. DF and TF conducted the AMS measurement and analysis with assistance from KW. AJH and JTHA developed the depth profile models. JTHA prepared the manuscript with contributions from all authors.

**Competing interests**

The authors declare that they have no conflict of interest.

**Acknowledgements**

We thank Craig Cary, Ian McDonald, Bob Dagg and Steph Lambie for assistance in the field, Antarctica New Zealand and Southern Lakes Helicopters for logistical support, and Steve Kotevski for laboratory assistance. We thank Jane Andersen and Greg Balco for their valuable reviews which improved the quality of the paper.

**Financial Support**

This research was supported by NZARI (RFP 2014-1), and ANSTO Portal grants 12215 and 12260 and an AINSE Postgraduate Research Award. JTHA was supported by a Sir Robin Irvine Scholarship, and a University of Otago departmental award. AA and ND were partially supported by the Russian Antarctic Expedition. We acknowledge the financial support from the Australian Government for the Centre for Accelerator Science at ANSTO through the National Collaborative Research Infrastructure Strategy (NCRIS). Prepared in part by LLNL under Contract DE-AC52-07NA27344; LDRD grant 19-LW-036. This is LLNL-JRNL-842669.

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
