# Peer review of "glaciers in the McMurdo Dry Valleys inferred from 10Be and 26Al"

_The Cryosphere, 2022_

## Referee Comment (RC1)

**Review of:** Antiphase dynamics between cold-based glaciers in the Antarctic Dry Valleys region and ice extent in the Ross Sea during MIS 5

by J. Anderson et al., The Cryosphere, March 2023

**General comments**

This is a very well written paper; I was especially impressed with the Introduction, 'Geologic setting and study area', and Methods sections. I find the potential antiphase dynamics between glacier and ice sheet extent very interesting and important to understand better. It is also fascinating that you find vertical mixing down to a depth of 70 cm in the Pearse valley depth profile. However, I have the following concerns that I think needs to be addressed before publication.

1. **Robustness of main result regarding antiphase dynamics.** You present $^{10}$Be-$^{26}$Al ages from only three samples of which one clearly has a large component of inheritance. I think you need to discuss the risk of inheritance in the remaining two samples and justify better why you trust these ages.
2. **Cosmogenic depth profile modelling.** I think this part of the manuscript is problematic for several reasons.
   a. Firstly, I think you need to demonstrate that you are aware that there is a difference in the expected cosmogenic profiles between scenarios of surface mixing with and without erosion (See Fig. 8 in Granger and Riebe, 2014; also shown in Fig. 4.24 in Dunai 2010). In your fig. 6D you show a scenario that looks comparable to mixing without erosion, although you suggest that your depth profiles are a result of both mixing and erosion. Depending on how long the sediment has been exposed, the erosion may not have gone to steady state, which would lead to an intermediate scenario between the two end-member scenarios shown in Granger and Riebe, 2014. I think you should illustrate and discuss this issue.
   b. Secondly, I am skeptical about your interpretation that the Pearse valley profile is a result of two depositional events. You seem to base this mostly on your cosmogenic profile modelling, without much evidence from the stratigraphy (except for the ice lenses). This is problematic because (i) your model does not include mixing (and therefore do not capture the result of combined mixing-and-erosion), (ii) modelling including mixing does a reasonably good job of matching most measured nuclide concentrations (See Figs. A and B below), and (iii) the inherited components of the measured cosmogenic nuclide concentrations have nearly the same concentrations and ratios (as shown in your Fig. 12a) – How likely is this outcome if there were two separate events? I think the simplest explanation is that there has been one depositional event leading to nearly (but not completely) homogenous initial nuclide concentrations that has subsequently undergone exposure, erosion and mixing. If you disagree, I think you need to further bolster your arguments for why this is not the case.
3. **Annual and inter-annual variability of active layer depth.** You assess the present-day active layer thickness based on the depth of ice-cemented sediments in your excavation (37 cm in Pearse Valley; a snapshot in time) and state that the ice-lenses

(73-86 cm in PV) represent a paleo-sublimation unconformity. I think you need to argue (based on literature) that this difference cannot be explained by annual/inter-annual variability of the active layer thickness, to justify why your interpreted 'paleosublimation unconformity' is not just a result of this variability.

[Figure]

*Figure A. Inverse modelling of the Pearse valley profile using a revised version of the model used in Andersen et al., 2018 and described in more detail in Knudsen et al., 2019. The top eight panels show the six model parameters (grey dots show burn-in phase and rejected models, colored dots show accepted models colored by walker number) and the pairwise trade-off between four of the sample parameters colored by the residual to the measured $^{10}Be$ and $^{26}Al$ concentrations (lower residual = better fit). Note that there is a trade-off between mixing depth (parameter '$d_m$' in eq. 4 in Knudsen et al. 2019 supplement) and mixing rate (m2/yr), with higher rates/lower depths leading to slightly better fits/lower residuals and correspondingly to a more abrupt 'step' in the profile between the mixed and unmixed zone. The three bottom panels show the measured $^{10}Be$ and $^{26}Al$ concentrations and the $^{26}Al/^{10}Be$ ratio as a function of depth with 1 sigma error bars (light grey samples with only one nuclide measured were not used for inversion). The grey patch/lines in background show accepted models and the red line the model with lowest residual/best fit to observations. No model matches all measured nuclide concentrations, but that is perhaps reasonable given the assumption about completely uniform inherited nuclide concentrations at t=0? This is not a perfect inversion and the production parameters are slightly different than in your manuscript, but based on this experiment I would encourage you to i) include mixing in your model and ii) reconsider why you think two-stage deposition is necessary to explain your nuclide concentrations.*

[Figure]

*Figure B. Same inversion as above, but here the top panels show the distributions of parameter values for accepted models and the trade-offs between surface exposure duration and inheritance while the lower panel shows the range in exhumation histories compatible with the measured nuclide concentrations.*

**Specific comments**

Section 4.2 and figures 8-10: I found the presentation of the modelling results in this section hard to follow. Firstly, if you decide to model several different scenarios, I think you need to make it much clearer in the text and figures when you are presenting and discussing what scenario. At present, the figure captions for example do not explain what model/scenario you are using, and there is no clear break in the text before you introduce your second scenario/model (the mid of line 450). It may be helpful to give the different scenarios descriptive names such as 'Scenario 1: Single depositional event, no vertical mixing' etc. Secondly, I don't think the link between ice lenses in the stratigraphy and your proposed breaks in the sedimentary sequence is convincingly presented. It would be helpful to have the

sedimentary 'logs' (Fig. 5) repeated next to the cosmogenic depth profiles to make it easier to compare, or alternatively indicate the key features (ice-lenses, transition between cemented and loose sediments) on/below/adjacent to the cosmogenic depth-profile. Thirdly, the top ice-lenses (~73-86 cm) are associated with the bottom of the (paleo-)active layer. If the lower set of ice-lenses (157-187 cm) indicate the bottom of a former active layer of comparable depth this would presumably imply ~0.5-1 m of erosion prior to deposition of the upper sediment package. This is not clear from the current description. Finally, I think you need to be clearer about how you assess the model performance, especially since you don't include mixing which inherently lead to a poor fit to the top (vertically mixed) samples.

I was intrigued to see that the cosmogenic profiles indicate rapid mixing down to ~70 cm depth in Pearse Valley. I would like to see a short discussion of what process can cause sediment mixing in these coarse-grained (not very frost-susceptible?) sediments in this climate?

L. 532-534: Could the 'offset' in 10Be concentrations be a result of mixing? Why do you not see an offset in 26Al-concentrations? Does this indicate that the measurement errors are underestimated?

**Technical corrections**

L. 82: Indicate location of Allan Hills BIA on map?

L. 92: Are the glacier advances in phase with ocean warming rather than out-of-phase?

L. 99-100: Can you add Arena and Kennar Valley on map?

L. 207-208: Add '(blue circle)' after first mention of drill site PV14-A and delete the second mention

L. 259-260: Did you also drill through the loose sand/gravel section at the top or did you dig?

L. 289-290 compared to L. 295-296. How can the sediments be ice-cemented but also loose? And would it be worthwhile indicating this (and the maybe also the laminae) on the sketch (Fig. 5)?

L. 293: You give an interpretation of the sediment deposition environment in Lower Wright valley, but not for Pearse Valley above.

L. 319: Check spelling of 'concentration'

L. 327-333: It is nice that you describe the error propagation in detail!

L. 373-374: What is the effect of ice lenses and the mix of open and ice-cemented porosity on this estimate?

Line 404: I would suggest that you illustrate the effect of mixing with/without erosion and add a panel F to this figure showing the resulting cosmogenic profiles corresponding to scenario E.

L. 419-420: Explain what you mean by 'simple' and 'complex' exposure history, this is not clear. Also, one sample indicate up to ~900 ka burial according to the diagram in Fig. 7, do you still consider this simple?

L. 433: Are the $^{10}$Be concentrations normalized by production rate (x-axes on Figs. 7 and 12)?

L. 443-444: How do you assess the model performance here? Do you calculate a residual?

L. 469: 'depositional age of the permafrost': permafrost is not deposited, sediments are?

L. 501: I don't think the lower panel is useful. Specify that you are talking about 'exposure age' on axis label.

L. 526-527. Here I think you need to discuss how certain you are about the present-day active layer thickness/on what time-scales it is expected to vary.

L. 533-534: Does an unconformity 'occur'?

L. 538-539: "The higher nuclide concentrations in these samples" is a bit misleading – the concentrations are lower than in the upper part of the section.

L. 590: spell out Acc. in caption and axes label.

L. 698: Check spelling of 'inheritance' in reference.

**References**

Andersen, J. L., Egholm, D. L., Knudsen, M. F., Linge, H., Jansen, J. D., Goodfellow, B. W., ... & Fredin, O. (2018). Pleistocene evolution of a Scandinavian plateau landscape. Journal of Geophysical Research: Earth Surface, 123(12), 3370-3387.

Dunai, T. J. (2010). *Cosmogenic nuclides: principles, concepts and applications in the earth surface sciences*. Cambridge University Press.

Granger, D. E., & Riebe, C. S. (2014). Cosmogenic nuclides in weathering and erosion. *Treatise on geochemistry 2nd edition*, 401-436.

Knudsen, M. F., Egholm, D. L., & Jansen, J. D. (2019). Time-integrating cosmogenic nuclide inventories under the influence of variable erosion, exposure, and sediment mixing. *Quaternary Geochronology*, *51*, 110-119.

---

## Author Comment (AC1)

**Response to reviewer #1 (Jane Andersen).**

We thank Jane Andersen for her detailed review and comments. We appreciate and acknowledge Jane's in-depth modelling of our data sets using a mixing depth model and the suggestions she provides based on her analyses.

Our responses to each comment are shown below in red.

Review of: Antiphase dynamics between cold-based glaciers in the Antarctic Dry Valleys region and ice extent in the Ross Sea during MIS 5

by J. Anderson et al., The Cryosphere, March 2023

**General comments**

This is a very well written paper; I was especially impressed with the Introduction, 'Geologic setting and study area', and Methods sections. I find the potential antiphase dynamics between glacier and ice sheet extent very interesting and important to understand better. It is also fascinating that you find vertical mixing down to a depth of 70 cm in the Pearse valley depth profile.

Thank you

However, I have the following concerns that I think needs to be addressed before publication.

1. Robustness of main result regarding antiphase dynamics. You present 10Be26Al ages from only three samples of which one clearly has a large component of inheritance. I think you need to discuss the risk of inheritance in the remaining two samples and justify better why you trust these ages.

The cobble sample PV14-CS3-P2 displays minimal edge rounding which suggests negligible erosion and is unlikely to be much older than the zero-erosion age. Given inheritance is stochastic, we infer that the two lowest consistent ages represent min inheritance and hence we take them to be our best estimate to represent the 'zero-erosion exposure ages' for the deposit of the cobbles that overlie the permafrost.

The $^{10}$Be concentrations of the cobbles (5 - 12 x $10^5$) are significantly lower – an order of magnitude – than the $^{10}$Be concentrations of the shallow/surface permafrost sediments (~4 x $10^6$). This gives us an additional piece of evidence that the cobble samples belong to a depositional event that post-dates the permafrost sediments and provides some confidence that the boulders are unlikely to have experienced a complex exposure history.

2. Cosmogenic depth profile modelling. I think this part of the manuscript is problematic for several reasons.

a. Firstly, I think you need to demonstrate that you are aware that there is a difference in the expected cosmogenic profiles between scenarios of surface mixing with and without erosion (See Fig. 8 in Granger and Riebe, 2014; also shown in Fig. 4.24 in Dunai 2010). In your fig. 6D you show a scenario that looks comparable to mixing without erosion, although you suggest that your depth profiles are a result of both mixing and erosion. Depending on how long the sediment has been exposed, the erosion may not have gone to steady state, which

would lead to an intermediate scenario between the two end-member scenarios shown in Granger and Riebe, 2014. I think you should illustrate and discuss this issue.

We agree that there may be a number of different scenarios from our depth profile. We will clarify this and include an intermediate scenario in Fig. 6 and in the text. Also please see our expanded and detailed reply to reviewer #2 (Greg Balco) with respect to depth profile modelling with and without a mixing term.

b. Secondly, I am skeptical about your interpretation that the Pearse valley profile is a result of two depositional events. You seem to base this mostly on your cosmogenic profile modelling, without much evidence from the stratigraphy (except for the ice lenses).

We interpret the two depositional events from the distinct offset at 2.09 m depth in the Pearse Valley profile. This is not based on depth profile modelling. We believe this offset is real and not due to analytical uncertainties or processes because both $^{26}$Al and $^{10}$Be show the same positive offset as a function of depth. (i.e., they represent an age-depth inversion reminiscent of a two-step depositional history).

This is problematic because (i) your model does not include mixing (and therefore do not capture the result of combined mixing-and-erosion), (ii) modelling including mixing does a reasonably good job of matching most measured nuclide concentrations (See Figs. A and B below), and (iii) the inherited components of the measured cosmogenic nuclide concentrations have nearly the same concentrations and ratios (as shown in your Fig. 12a) –

First, we again thank Jane (reviewer #1) for doing depth profile analyses of our data using her model.

Second, we agree that our data analysis and depth profile modelling does not include an active-layer mixing component which as Jane points out, has the potential to better constrain the evolution of the permafrost core. With that said, we are not convinced it is necessary to include a mixing model (see our detailed reply to the same comment made by reviewer #2 (Greg Balco). As noted by Greg, and to a degree by Jane in her review here, the mixing model of Andersen / Knudsen does not result in a conflicting or challenging interpretation of the data from what we arrive at using a non-mixing depth profile model of Hidy et al. (2018). We note here our detailed reply to reviewer #2 on this item.

In summary, introducing a mixing model to our depth profiles we believe does not improve the age-inheritance-erosion constraints we arrive at nor resolve the complexity of these data, and hence does not alter our conclusions.

How likely is this outcome if there were two separate events? I think the simplest explanation is that there has been one depositional event leading to nearly (but not completely) homogenous initial nuclide concentrations that has subsequently undergone exposure, erosion and mixing. If you disagree, I think you need to further bolster your arguments for why this is not the case.

We agree that the initial state is a well-mixed sediment profile – the corollary is a constant inheritance component at all depths. On the assumption that all samples have maintained their relative depths to each other, a single depositional event is the most likely and simplest outcome. However, given a distinct increase in nuclide concentration at 2.09 m depth for both $^{10}$Be and $^{26}$Al, relative to the samples between 1.09 – 1.65 m depth, and the presence of ice lenses between 1.57 – 1.87 m depth, we chose to interpret the deeper core section to signal an earlier depositional event. We will provide further clarification of this in the text.

3. Annual and inter-annual variability of active layer depth. You assess the present-day active layer thickness based on the depth of ice-cemented sediments in your excavation (37 cm in Pearse Valley; a snapshot in time) and state that the ice-lenses (73-86 cm in PV) represent a paleo-sublimation unconformity. I think you need to argue (based on literature) that this difference cannot be explained by annual/interannual variability of the active layer thickness, to justify why your interpreted 'paleosublimation unconformity' is not just a result of this variability.

We cannot rule out that the paleosublimation unconformity is a result of a warm summer. However, most of the coastal thaw zone rarely exceeds >50 cm depth of the active layer in Taylor Valley. As Pearse Valley is further inland and at higher elevation this seems less likely. We will acknowledge the possibility of this in the text.

Figure A. Inverse modelling of the Pearse valley profile using a revised version of the model used in Andersen et al., 2018 and described in more detail in Knudsen et al., 2019. The top eight panels show the six model parameters (grey dots show burn-in phase and rejected models, colored dots show accepted models colored by walker number) and the pairwise tradeoff between four of the sample parameters colored by the residual to the measured 10Be and 26Al concentrations (lower residual = better fit). Note that there is a trade-off between mixing depth (parameter 'dm' in eq. 4 in Knudsen et al. 2019 supplement) and mixing rate (m2/yr), with higher rates/lower depths leading to slightly better fits/lower residuals and correspondingly to a more abrupt 'step' in the profile between the mixed and unmixed zone. The three bottom panels show the measured 10Be and 26Al concentrations and the 26Al/10Be ratio as a function of depth with 1 sigma error bars (light grey samples with only one nuclide measured were not used for inversion). The grey patch/lines in background show accepted models and the red line the model with lowest residual/best fit to observations. No model matches all measured nuclide concentrations, but that is perhaps reasonable given the assumption about completely uniform inherited nuclide concentrations at t=0? This is not a perfect inversion and the production parameters are slightly different than in your manuscript, but based on this experiment I would encourage you to i) include mixing in your model and ii) reconsider why you think two-stage deposition is necessary to explain your nuclide concentrations.

Figure B. Same inversion as above, but here the top panels show the distributions of parameter values for accepted models and the trade-offs between surface exposure duration and inheritance while the lower panel shows the range in exhumation histories compatible with the measured nuclide concentrations.

See our detailed replies above and to reviewer #2.

**Specific comments**

Section 4.2 and figures 8-10: I found the presentation of the modelling results in this section hard to follow. Firstly, if you decide to model several different scenarios, I think you need to make it much clearer in the text and figures when you are presenting and discussing what scenario. At present, the figure captions for example do not explain what model/scenario you are using, and there is no clear break in the text before you introduce your second scenario/model (the mid of line 450). It may be helpful to give the different scenarios descriptive names such as 'Scenario 1: Single depositional event, no vertical mixing' etc.

We will label and explain scenarios more clearly in the figures and text.

Secondly, I don't think the link between ice lenses in the stratigraphy and your proposed breaks in the sedimentary sequence is convincingly presented. It would be helpful to have

the sedimentary 'logs' (Fig. 5) repeated next to the cosmogenic depth profiles to make it easier to compare, or alternatively indicate the key features (ice-lenses, transition between cemented and loose sediments) on/below/adjacent to the cosmogenic depth-profile.

We will show the logs and key features next to the depth profile data.

Thirdly, the top icelenses (~73-86 cm) are associated with the bottom of the (paleo-)active layer. If the lower set of ice-lenses (157-187 cm) indicate the bottom of a former active layer of comparable depth this would presumably imply ~0.5-1 m of erosion prior to deposition of the upper sediment package. This is not clear from the current description.

In the text we do not suggest the lower set of ice lenses are necessarily a paleosublimation unconformity like the top ice lenses. We will include this possible scenario in the text and discuss such an erosion scenario.

Finally, I think you need to be clearer about how you assess the model performance, especially since you don't include mixing which inherently lead to a poor fit to the top (vertically mixed) samples.

We will clarify this in the text. We will provide further detail assessing the model performance.

I was intrigued to see that the cosmogenic profiles indicate rapid mixing down to ~70 cm depth in Pearse Valley. I would like to see a short discussion of what process can cause sediment mixing in these coarse-grained (not very frost-susceptible?) sediments in this climate?

The active-layer is susceptible to cryoturbation during summer months. Gravimetric water content is relatively high in near-surface permafrost in the Dry Valleys. See Fig 2e in Lacelle et al. (2022). We will include a short discussion about these processes.

L. 532-534: Could the 'offset' in 10Be concentrations be a result of mixing? Why do you not see an offset in 26Al-concentrations? Does this indicate that the measurement errors are underestimated?

The offset is a result of mixing. The paleoactive-layer above this depth was vertically mixed. We will clarify this in the text.

Regarding the offset in $^{26}$Al concentrations, the averaging of the first five data points (the average mixed layer concentration) in both profiles up to and including 0.65 m, is distinctly different for the next lower data point below the 0.65 m depth.

**Technical corrections**

L. 82: Indicate location of Allan Hills BIA on map? Agree

L. 92: Are the glacier advances in phase with ocean warming rather than out-of-phase? We will correct this

L. 99-100: Can you add Arena and Kennar Valley on map? Agree

L. 207-208: Add '(blue circle)' after first mention of drill site PV14-A and delete the second mention Agree

L. 259-260: Did you also drill through the loose sand/gravel section at the top or did you dig? The top of the section was collected via a mixture of digging and coring. The initial coring

was collected in whirl-pak bags as the coring quality was poor, and largely came out as loose material. We will clarify this.

L. 289-290 compared to L. 295-296. How can the sediments be ice-cemented but also loose? And would it be worthwhile indicating this (and the maybe also the laminae) on the sketch (Fig. 5)? The active-layer (0 – 28 cm) above the ice cemented permafrost consists of a thin armoured surface layer (2 cm) and a layer of loose sand and pebbles (26 cm). Below 28 cm is ice-cemented. We will clarify the sentence so it is easier to follow. This is already outlined from the key in Fig. 5. We will include laminae in Fig. 5.

L. 293: You give an interpretation of the sediment deposition environment in Lower Wright valley, but not for Pearse Valley above. We will include our interpretation for Pearse Valley.

L. 319: Check spelling of 'concentration' Agree

L. 327-333: It is nice that you describe the error propagation in detail! Thank you

L. 373-374: What is the effect of ice lenses and the mix of open and ice-cemented porosity on this estimate? The difference between bulk density for loose sediment, and ice cemented-permafrost is largely within the +/- 0.1 uncertainty. All ice lenses were less than 10 cm thick and, in most cases, less than 5 cm thick. We will acknowledge the small difference this assumption could have on the overall models.

Line 404: I would suggest that you illustrate the effect of mixing with/without erosion and add a panel F to this figure showing the resulting cosmogenic profiles corresponding to scenario E. We will add a panel F showing erosion.

L. 419-420: Explain what you mean by 'simple' and 'complex' exposure history, this is not clear. Also, one sample indicate up to ~900 ka burial according to the diagram in Fig. 7, do you still consider this simple? Simple assumes a simple constant exposure. Complex suggests at least one episode of burial. We will clarify this in the text. The sample showing up to ~900 ka burial has a higher probability of being a non-simple exospore history. We will distinguish between that sample and the other two in the revised manuscript.

L. 433: Are the 10Be concentrations normalized by production rate (x-axes on Figs. 7 and 12)?

No, our concentrations are not normalised. We will clarify this in the text.

L. 443-444: How do you assess the model performance here? Do you calculate a residual?

We didn't calculate residuals, but from the model outputs the model fits within the 1-sigma measurement error bars for each sample within the deeper depth range, and they do not fit within 1-sigma for the shallow depth range. We will include a calculation of the residuals in the revised version over the two depth ranges to support this statement more quantitatively.

L. 469: 'depositional age of the permafrost': permafrost is not deposited, sediments are? Agree. This will be corrected to 'depositional age of the sediment'

L. 501: I don't think the lower panel is useful. Specify that you are talking about 'exposure age' on axis label. Exposure age will be shown on both plots

L. 526-527. Here I think you need to discuss how certain you are about the present-day active layer thickness/on what time-scales it is expected to vary. The present-day active layer fluctuates throughout summer months. We will include further discussion regarding active layer migration over summer months.

L. 533-534: Does an unconformity 'occur'? No we will change this to 'formed'

L. 538-539: "The higher nuclide concentrations in these samples" is a bit misleading – the concentrations are lower than in the upper part of the section. We will clarify this sentence to state: The increase in nuclide concentration >2.09 m depth relative to the samples between 1.09 - 1.65 m depth.

L. 590: spell out Acc. in caption and axes label. Agree

L. 698: Check spelling of 'inheritance' in reference. Agree

---

## Author Response (AR1)

**Response to reviewer #1 (Jane Andersen).**

We thank Jane Andersen for her detailed review and comments. We appreciate and acknowledge Jane's in-depth modelling of our data sets using a mixing depth model and the suggestions she provides based on her analyses.

Our responses to each comment are shown below in red.

Review of: Antiphase dynamics between cold-based glaciers in the Antarctic Dry Valleys region and ice extent in the Ross Sea during MIS 5

by J. Anderson et al., The Cryosphere, March 2023

**General comments**

This is a very well written paper; I was especially impressed with the Introduction, 'Geologic setting and study area', and Methods sections. I find the potential antiphase dynamics between glacier and ice sheet extent very interesting and important to understand better. It is also fascinating that you find vertical mixing down to a depth of 70 cm in the Pearse valley depth profile.

Thank you

However, I have the following concerns that I think needs to be addressed before publication.

1. Robustness of main result regarding antiphase dynamics. You present 10Be26Al ages from only three samples of which one clearly has a large component of inheritance. I think you need to discuss the risk of inheritance in the remaining two samples and justify better why you trust these ages.

The cobble sample PV14-CS3-P2 displays minimal edge rounding which suggests negligible erosion and is unlikely to be much older than the zero-erosion age. Given inheritance is stochastic, we infer that the two lowest consistent ages represent min inheritance and hence we take them to be our best estimate to represent the 'zero-erosion exposure ages' for the deposit of the cobbles that overlie the permafrost. See additional discussion on lines 531 – 533 (Sect. 4.5).

The $^{10}$Be concentrations of the cobbles ($5 - 12 \times 10^5$) are significantly lower – an order of magnitude – than the $^{10}$Be concentrations of the shallow/surface permafrost sediments ($\sim 4 \times 10^6$). This gives us an additional piece of evidence that the cobble samples belong to a depositional event that post-dates the permafrost sediments and provides some confidence that the boulders are unlikely to have experienced a complex exposure history.

2. Cosmogenic depth profile modelling. I think this part of the manuscript is problematic for several reasons.

a. Firstly, I think you need to demonstrate that you are aware that there is a difference in the expected cosmogenic profiles between scenarios of surface mixing with and without erosion (See Fig. 8 in Granger and Riebe, 2014; also shown in Fig. 4.24 in Dunai 2010). In your fig. 6D you show a scenario that looks comparable to mixing without erosion, although you suggest that your depth profiles are a result of both mixing and erosion. Depending on how long the sediment has been exposed, the erosion may not have gone to steady state, which would lead to an intermediate scenario between the two end-member scenarios shown in Granger and Riebe, 2014. I think you should illustrate and discuss this issue.

We agree that there may be a number of different scenarios from our depth profile. We have updated Fig. 6 (now Fig. 7) which now includes vertical mixing via active-layer cryoturbation with no erosion (Fig. 7d) and with steady-state erosion (Fig. 7e).

b. Secondly, I am skeptical about your interpretation that the Pearse valley profile is a result of two depositional events. You seem to base this mostly on your cosmogenic profile modelling, without much evidence from the stratigraphy (except for the ice lenses).

We interpret the two depositional events from the offset for both [10]Be and [26]Al at 2.09 m depth in the Pearse Valley profile. This is not based on depth profile modelling.

This is problematic because (i) your model does not include mixing (and therefore do not capture the result of combined mixing-and-erosion), (ii) modelling including mixing does a reasonably good job of matching most measured nuclide concentrations (See Figs. A and B below), and (iii) the inherited components of the measured cosmogenic nuclide concentrations have nearly the same concentrations and ratios (as shown in your Fig. 12a) –

We are not convinced it is necessary to include a mixing model, though we acknowledge it is a possible interpretation among many possible scenarios. We discuss these limitations in Section 4.1 and 5.1.

As reviewer one's (JA's) model demonstrates, it does not add much useful constraint on age or erosion rate. In vertically mixed soils (such as those in the Dry Valleys), the average production rate in the soil is constant throughout the active-layer (mixed zone) (Granger and Riebe, 2014). Under these conditions, a large sample of grains in permafrost soils reflect the spatially averaged cosmogenic nuclide concentration from the deposit. When the residence time (i.e., depositional age) of the quartz grains within the mixed zone is significantly less than the half-life of the cosmogenic nuclide (for 10Be, λ = 1.39 Ma), losses due to radioactive decay can be safely ignored. Therefore, assuming the quartz grains in a given deposit are well mixed, inherited nuclides would be the same at all depths in the mixed layer. This can be applied to the upper 65 cm without a mixing model. See lines 591 - 596

Reviewer two (GB) also pointed out JA's / Knudsen's model still doesn't include episodic ice-cover effect, which could vary age, and erosion results more than the wide-range of reviewer one's model output.

Thus, introducing a mixing model to our depth profiles does not solve the complexity of these data, nor change our results or conclusions. Instead, we include GB's suggestion using a simple calculation for the upper 65 cm mixing zone. See section 4.2.

How likely is this outcome if there were two separate events? I think the simplest explanation is that there has been one depositional event leading to nearly (but not completely) homogenous initial nuclide concentrations that has subsequently undergone exposure, erosion and mixing. If you disagree, I think you need to further bolster your arguments for why this is not the case.

We acknowledge one depositional event is a possible scenario. However, we prefer the two depositional events scenario from the distinct increase in nuclide concentration at 2.09 m depth for both [10]Be and [26]Al, relative to the samples between 1.09 – 1.65 m depth. This observation corresponds with the presence of the ice lenses between 1.57 – 1.87 m depth. In section 4.3 we provide further information explaining why we prefer the two depositional events.

3. Annual and inter-annual variability of active layer depth. You assess the present-day active layer thickness based on the depth of ice-cemented sediments in your excavation (37 cm in Pearse Valley; a snapshot in time) and state that the ice-lenses (73-86 cm in PV) represent a paleo-sublimation unconformity. I think you need to argue (based on literature) that this difference cannot be explained by annual/interannual variability of the active layer thickness, to justify why your interpreted 'paleosublimation unconformity' is not just a result of this variability.

We cannot rule out that the paleosublimation unconformity is a result of a warm summer. However, most of the coastal thaw zone rarely exceeds >50 cm depth of the active layer in Taylor Valley. As Pearse Valley is further inland and at higher elevation this seems less likely. We acknowledge the possibility on lines 570 – 575.

Figure A. Inverse modelling of the Pearse valley profile using a revised version of the model used in Andersen et al., 2018 and described in more detail in Knudsen et al., 2019. The top eight panels show the six model parameters (grey dots show burn-in phase and rejected models, colored dots show accepted models colored by walker number) and the pairwise tradeoff between four of the sample parameters colored by the residual to the measured 10Be and 26Al concentrations (lower residual =

better fit). Note that there is a trade-off between mixing depth (parameter 'dm' in eq. 4 in Knudsen et al. 2019 supplement) and mixing rate (m2/yr), with higher rates/lower depths leading to slightly better fits/lower residuals and correspondingly to a more abrupt 'step' in the profile between the mixed and unmixed zone. The three bottom panels show the measured 10Be and 26Al concentrations and the 26Al/10Be ratio as a function of depth with 1 sigma error bars (light grey samples with only one nuclide measured were not used for inversion). The grey patch/lines in background show accepted models and the red line the model with lowest residual/best fit to observations. No model matches all measured nuclide concentrations, but that is perhaps reasonable given the assumption about completely uniform inherited nuclide concentrations at t=0? This is not a perfect inversion and the production parameters are slightly different than in your manuscript, but based on this experiment I would encourage you to i) include mixing in your model and ii) reconsider why you think two-stage deposition is necessary to explain your nuclide concentrations.

Figure B. Same inversion as above, but here the top panels show the distributions of parameter values for accepted models and the trade-offs between surface exposure duration and inheritance while the lower panel shows the range in exhumation histories compatible with the measured nuclide concentrations.

**Specific comments**

Section 4.2 and figures 8-10: I found the presentation of the modelling results in this section hard to follow. Firstly, if you decide to model several different scenarios, I think you need to make it much clearer in the text and figures when you are presenting and discussing what scenario. At present, the figure captions for example do not explain what model/scenario you are using, and there is no clear break in the text before you introduce your second scenario/model (the mid of line 450). It may be helpful to give the different scenarios descriptive names such as 'Scenario 1: Single depositional event, no vertical mixing' etc.

We have labelled and explained model scenarios more clearly in the figures and text. See Section 4.2 and 4.3, and Fig. 8.

Secondly, I don't think the link between ice lenses in the stratigraphy and your proposed breaks in the sedimentary sequence is convincingly presented. It would be helpful to have the sedimentary 'logs' (Fig. 5) repeated next to the cosmogenic depth profiles to make it easier to compare, or alternatively indicate the key features (ice-lenses, transition between cemented and loose sediments) on/below/adjacent to the cosmogenic depth-profile.

Figs 4, 5, 8 and 10 now show the logs and key features next to the depth profile data.

Thirdly, the top icelenses (~73-86 cm) are associated with the bottom of the (paleo-)active layer. If the lower set of ice-lenses (157-187 cm) indicate the bottom of a former active layer of comparable depth this would presumably imply ~0.5-1 m of erosion prior to deposition of the upper sediment package. This is not clear from the current description.

In the text we do not suggest the lower set of ice lenses are necessarily a paleosublimation unconformity like the top ice lenses. We include this possible scenario in the text. See lines 581 – 583.

Finally, I think you need to be clearer about how you assess the model performance, especially since you don't include mixing which inherently lead to a poor fit to the top (vertically mixed) samples.

See updated text in Sections 4 and 5 discussing surface mixing. We provide further detail assessing the model performance including residual. See lines 473 – 478 and SD3.

I was intrigued to see that the cosmogenic profiles indicate rapid mixing down to ~70 cm depth in Pearse Valley. I would like to see a short discussion of what process can cause sediment mixing in these coarse-grained (not very frost-susceptible?) sediments in this climate?

The active-layer is susceptible to cryoturbation during summer months. Gravimetric water content is relatively high in near-surface permafrost in the Dry Valleys which can result in cryoturbation. See Fig 2e in Lacelle et al. (2022). We include a short discussion about these processes. See lines 573 – 575.

L. 532-534: Could the 'offset' in 10Be concentrations be a result of mixing? Why do you not see an offset in 26Al-concentrations? Does this indicate that the measurement errors are underestimated?

The offset is a result of mixing. The paleoactive-layer above this depth was vertically mixed. Except for sample PV14-A-02, all 26Al-concentrations are ~1.85 – 1.95 x $10^7$ and show an offset. We acknowledge there is more scatter than the 10Be offset.

**Technical corrections**

L. 82: Indicate location of Allan Hills BIA on map? Added

L. 92: Are the glacier advances in phase with ocean warming rather than out-of-phase? The Dry Valleys glacier advances are out of phase with ice advance in the Ross Sea. They are out of phase with marine based ice sheet retreat.

L. 99-100: Can you add Arena and Kennar Valley on map? Added

L. 207-208: Add '(blue circle)' after first mention of drill site PV14-A and delete the second mention Updated

L. 259-260: Did you also drill through the loose sand/gravel section at the top or did you dig? The top of the section was collected via a mixture of digging and coring. The initial coring was collected in whirl-pak bags as the coring quality was poor, and largely came out as loose material. See 236 – 238.

L. 289-290 compared to L. 295-296. How can the sediments be ice-cemented but also loose? And would it be worthwhile indicating this (and the maybe also the laminae) on the sketch (Fig. 5)? The active-layer (0 – 28 cm) above the ice cemented permafrost consists of a thin armoured surface layer (2 cm) and a layer of loose sand and pebbles (26 cm). Below 28 cm is ice-cemented. See line 270 – 272.

L. 293: You give an interpretation of the sediment deposition environment in Lower Wright valley, but not for Pearse Valley above. We have updated. See line 253.

L. 319: Check spelling of 'concentration' Updated

L. 327-333: It is nice that you describe the error propagation in detail! Thank you

L. 373-374: What is the effect of ice lenses and the mix of open and ice-cemented porosity on this estimate? The difference between bulk density for loose sediment, and ice cemented-permafrost is largely within the +/- 0.1 uncertainty. All ice lenses were less than 10 cm thick and, in most cases, less than 5 cm thick. We acknowledge the small difference this assumption could have on the overall models on lines 372 – 374.

Line 404: I would suggest that you illustrate the effect of mixing with/without erosion and add a panel F to this figure showing the resulting cosmogenic profiles corresponding to scenario E. We have added the additional panel. See Fig. 7.

L. 419-420: Explain what you mean by 'simple' and 'complex' exposure history, this is not clear. Also, one sample indicate up to ~900 ka burial according to the diagram in Fig. 7, do you still consider this simple? Simple assumes a simple constant exposure. Complex suggests at least one episode of burial. The sample with up to ~900 ka burial has an exceptionally large uncertainty. As explained in the updated text, we infer the two lowest consistent ages represent the minimum inheritance.

L. 433: Are the 10Be concentrations normalized by production rate (x-axes on Figs. 7 and 12)?

No, our concentrations are not normalised. We have modified the fig captions and provide additional context at the end of Sect. 5.2

L. 443-444: How do you assess the model performance here? Do you calculate a residual?

We calculated a residual for both the full depth profile (0.02 – 3.16 m) and the upper sediment samples (0.02 – 1.65 m) see lines 473 – 477 and SD3.

L. 469: 'depositional age of the permafrost': permafrost is not deposited, sediments are? Agree. This is corrected to deposition age through the text.

L. 501: I don't think the lower panel is useful. Specify that you are talking about 'exposure age' on axis label. Both axis now show 'exposure age'.

L. 526-527. Here I think you need to discuss how certain you are about the present-day active layer thickness/on what time-scales it is expected to vary. The present-day active layer fluctuates throughout summer months. We include further discussion regarding active layer migration over summer months. See lines 569 – 573.

L. 533-534: Does an unconformity 'occur'? No, we changed this to 'formed'. See line 568.

L. 538-539: "The higher nuclide concentrations in these samples" is a bit misleading – the concentrations are lower than in the upper part of the section. We will clarify this sentence to state: The increase in nuclide concentration >2.09 m depth relative to the samples between 1.09 - 1.65 m depth. See line 579.

L. 590: spell out Acc. in caption and axes label. This has been modified on Fig. 13

L. 698: Check spelling of 'inheritance' in reference. This has been corrected.

**Response to reviewer #2 (Greg Balco).**

We thank Greg Balco for his detailed review and comments. Our responses to each comment are shown below in red.

The summary of this review is that this is a straightforward paper that reports a collection of useful cosmogenic-nuclide data from the Antarctic Dry Valleys. From this perspective the paper is perfectly fine and communicates these data in a clear way. Thank you

There is one thing that is surprising about this paper, though, which is that the clear description of the cosmogenic-nuclide data is bookended by an extraordinarily large amount of discussion of the relationship of glacier advances to climate in the DV. The proposed relationships are very likely to be true, but are almost entirely unrelated to the new observations in this paper. To put this in perspective, this paper reports cosmogenic-nuclide measurements on 27 samples -- three surface samples and the remainder from cores in frozen sediment. Of the 3 surface samples, only two have the same age. These two ages are consistent with the generally accepted and well-described-in-the-literature concept that Dry Valleys glaciers are moisture-starved and retreat during cold periods. Thus, 2/27 (7.5%) of the data described in this paper can be used to support this assertion. However, 100% of the title, about 2/3 of the abstract, most of the introduction, a very long discussion in section 5.4, and about 2/3 of the conclusions are devoted to the antiphase-with-climate behaviour of Dry Valleys glaciers. Thus, nearly all of the introduction, discussion, and conclusions of the paper is all about only 7.5% of the data.

We appreciate the remark of the imbalance in the direction of the paper with respect to sample inventory and accept the need to address it. See below.

The data from the other 92.5% of the samples, which are depth profile measurements, are quite interesting, but complicated and not easily interpretable as an age of any particular event. They are

certainly consistent with glaciers retreating during glacial maxima, but it is clear that the measured nuclide concentrations record a complex history of exposure, burial, sediment recycling, and permafrost processes, so they would be equally well consistent with all kinds of other glacier change scenarios. These data are not really about glacier change, they are more about the provenance and process environment of frozen sediments in the DV. It appears that the authors thought that these data were too complicated, and permafrost dynamics too esoteric, for anyone to care about, so they wrote the entire paper about the two surface samples and then included the depth profiles because they were from about the same place.

In our response, we note that considerable effort was invested in modelling the permafrost depth profiles and the relevancy of the data to glacial and sedimentary processes in the Dry Valleys. We expand on our interpretation and significance of the depth profiles throughout the revised manuscript.

This is not to say that there's not a good description of the depth profile data in this paper -- there is -- but it is just kind of abandoned among all the discussion of antiphase glacier dynamics, which, as noted, is certainly correct, but already pretty well established by other research and only marginally relevant to the majority of the data reported here.

Overall, we agree with the reviewers' comments with respect to most data presented not being directly related to glacier advance / retreat. We have adjusted the title, abstract, introduction, results, discussion, and conclusions to include greater emphasis on permafrost processes, and sediment recycling.

Of course if I were writing this I would have titled it "Timescale of active layer mixing in Antarctic permafrost inferred from cosmogenic-nuclide depth profiles," spent most of the paper talking about that, and then just tossed in a sentence at the end that, oh yeah, we measured three surface clasts, of which two ages agree and are consistent with the generally accepted thinking that DV glaciers retreated during the LGM.

We have modified the title to reflect both permafrost processes and antiphase dynamics of Taylor Glacier.

New title: *Antarctic permafrost processes and antiphase dynamics of cold-based glaciers in the McMurdo Dry Valleys inferred from $^{10}$Be and $^{26}$Al cosmogenic-nuclides.*

But, I'm not writing this, so the authors can do whatever they want. All the data are clearly reported here, so readers can take from it what they choose.

However, even though the authors don't seem very interested in highlighting the depth-profile results, I am going to only talk about them in the rest of the review. Basically, what happens here is that the data clearly show that the concentration-depth relationship does not show the exponential-like decrease expected for a sediment/soil unit that is either stable or eroding without vertical mixing. A surface mixed layer is clearly evident in both profiles. However, despite this observation, the authors begin by trying to fit the data with a model that does not include a mixed layer (Hidy, various refs.). Frankly, this does not make a ton of sense. Why is this even included? Eventually at the end of this section, the authors come to the conclusion that "the depth profile model does not work well for non-attenuating profiles." Of course it doesn't! It's not supposed to. Thus, I would remove this entire initial fitting exercise.

We presented the initial model to demonstrate, as pointed out by the reviewer, that it does not fit our data – as one would expect - given there is a surface mixed layer. However, it does provide a reasonable or perhaps a sufficient fit at mid to lower depths and as such gives some direction in interpreting the evolution of the permafrost deposit.

We have re-written the text in this Section 4 and 5 and highlight the limitations in depth models for complex data sets as an instructive exercise for those less familiar in this field.

In the next fitting exercise, the authors average all the data from the apparent mixed layer into one mixed pseudo-sample, and then try to fit the same model. As expected, this works better, but this is also a surprising thing to do, because, basically, you still have a data set that shows clearly that the

process of mixing is happening, but you are trying to fit it with a model that doesn't include that process. Of course you can kind of fit it with this approach, but it's not right because it doesn't capture the effect that the production rate in the entire mixed layer is not the same as the production rate at the average depth in the mixed layer, or the effect that mixing and erosion are happening at the same time, if that makes any sense. So, this is a little better in terms of fitting the data, but kind of what has happened here is the authors have tortured the data to try to fit them with a model they know to be wrong. This could be done better.

The other review (Jane Lund Anderson) discusses this at length and applies a much more sophisticated model to fit the data. As expected, this does a much better job of fitting the data and also highlights the tradeoff between various parameters (like exposure age and inheritance) that makes it basically impossible to assign a definitive age to this deposit based on the depth-profile data. It should also be noted, however, that the results of this model simulation don't actually account for the fact that the site in Pearse Valley is episodically covered by ice, and, presumably, vertical mixing also stops during these periods. So even the large range of exposure ages permitted by these results might not cover all possible scenarios. As an aside that's not really related to this paper, the Andersen/Knudsen model doesn't do a good job of reproducing the sharp bottom of a mixed layer that is often observed in data (although it's not designed to).

As we outlined in the response to reviewer #1 -We are not convinced it is necessary to include a mixing model, nor do we believe that in this case, it solves the problem uniquely, or even sufficiently, for age, inheritance, and surface erosion. As reviewer #1's (JA's) model demonstrates, and as repeated by this reviewer, it appears not to result in a more restricted constraint on age or erosion rate.

We are not convinced it is necessary to include a mixing model, though we acknowledge it is a possible interpretation among many possible scenarios. We discuss these limitations in Section 4.1 and 5.1.

We acknowledge a mixing model could provide an improved fit to our data among many possible scenarios, but not necessarily an improved interpretation due to the complexity of the data set and uncertainties in age and erosion rate solutions. We expand upon this in our discussion, importantly, in this case, not as one contradictory or as a challenge to the one we have obtained for the Pearse Valley permafrost core in the absence of a mixing model.

In continuously vertically mixed soils where radionuclide decay is negligible (such as those in the Dry Valleys), the average production rate in the mixed layer is constant with depth (Granger and Riebe, 2014). Under these conditions, the mixed layer in our permafrost soils will reflect the spatially averaged cosmogenic nuclide concentration. Surface erosion or changes in active layer thickness will modify this outcome, however the Antarctic environment supports extremely low erosion rates and hence the age-erosion trade off as mentioned by reviewer #2 may not be severe. In summary assuming the quartz grains in a given deposit are well mixed, on a time scale shorter than the $^{26}Al$ half-life, the steady state final nuclide concentration should be constant at all depths which is what we observe in the surface mixed layer at Pearse Valley. Thus, a depth model fit, without a mixing component, commencing with a calculated mean concentration at mid-depth within the upper 65 cm i.e., the surface mixed layer provides a reasonable attempt at data analysis.

In addition, we note that reviewer #2 (GB) highlighted the fact that the mixing model output of reviewer #1's (JA) model does not include the effect of episodic ice-cover, which could vary age, and erosion results more than the wide-range of reviewer #1's model output.

Overall, however, the important thing is that even though the more sophisticated model provides an age estimate, it's also dependent on assumptions about what happened re. ice cover, etc. So the fact is that these data are not going to provide an accurate age estimate for any particular event. We agree with this, though the data can still provide useful age information.

One thing that would have intermediate complexity and possibly be helpful in interpreting these data would be to try to fit the data with the mixed layer model of Lal and Chen (2005). This is fairly simplified (the parameters are exposure time, erosion rate, and mixed layer depth), but it would be helpful to see what range of erosion rates and exposure times could be consistent with the data. Of

course this doesn't include episodic burial by ice either. If the authors try this they should note that there is a typo in Equation 12 of Lal and Chen -- it is missing a factor of rho in one of the terms.

We agree there are many different scenarios. As this doesn't include the episodic burial by ice it also does not solve the complexity of this data and is unlikely to change our results or conclusions.

I am also skeptical of the "paleosublimation unconformity." Certainly this could be true, but there is no strong evidence that it is. See additional discussion why a paleosublimation seems plausible on lines 569 – 573.

What these data do provide, however, is some interesting information about soil/sediment mixing in these areas, which is something that is not widely studied or understood.
Thank you. We agree that this data set provides an interesting perspective on permafrost sediments and processes that are not well understood. We emphasize this on lines 50 – 53 and throughout the updated manuscript.

Like the first reviewer, I am really interested to see very thorough mixing in the upper 70 cm in the Pearse Valley core. That's not necessarily expected. So in my view these data are extremely unhelpful as to age, but quite interesting as to process. If I were writing this, I'd probably do this as follows:

As we noted above, we agree with the reviewers' comments with respect to the majority of data presented not being directly related to glacier advance / retreat and will expand on permafrost processes, and sediment recycling. We agree that the entire profile may not be particularly useful for accurate age derivation, but we stand by our observation that depth concentration offsets do demonstrate several deposition / mixing events, which can be associated with relative age, and therefore permafrost stability.

1. Note the main features of the Pearse Valley depth profile data, which are (i) there is a surface mixed layer; (ii) there is a high inherited concentration at depth, and (iii) the 26/10 ratio indicates a long history of exposure and burial. See revised Sections 4 and 5.

2. Note the main feature of the Wright Valley profile, which is that it has the same concentration at all depths. Noted in Sect 4

3. Apply a simple calculation to determine the "age" of the mixed layer at Pearse by (i) computing excess Be-10 and Al-26 with respect to the inheritance at the bottom of the cores, and (ii) dividing by the production rate to get an age. For Pearse Valley, there is about 5e5 atoms/g excess Be-10 in the mixed layer, and the production rate in the mixed layer looks to be about 5 atoms/g/yr, which suggests that order 100 ka is needed to produce these data. Looks like you get a similar order of magnitude from the Al-26 data.
We include reviewer #2's suggestion using a simple 'back-of-the-envelope' calculation (ie (min-max)/(average production rate)) for the upper 65 cm mixed zone. See Sect 4.2.

4. Apply another simple calculation to estimate the length of the minimum total exposure plus burial history recorded by the inherited 26-10 concentrations. This will come out to be 1 Ma++ and shows that these sediments have been sitting here for a long time.
See Sect. 5.2

5. Apply another simple calculation at Wright Valley to show that only a very short period of exposure can have taken place after the sediments were mixed/emplaced, which is already in line 264 (< LGM).
This is included.
The overall conclusions being that the Pearse Valley sediments have been there for a while, but are subject to fairly rapid active-layer mixing, at least some of the time, whereas th Wright Valley sediments don't seem to have been there for very long.
The text currently notes the >1 Ma scale of pre-depositional history of both the Pearse Valley and Wright Valley sediments in their respective cores (see Sect 5.2). We have restricted our discussion following the steps of reviewer #2.

I am not sure you can get much else out of these data. They are interesting from the process perspective, but I don't think they do much from the chronology perspective. In this paper, the emphasis on trying to come up with an age in order to fit these data into the overall discussion of glacier change isn't really a good fit and is not highly informative.

To summarize, I don't really have any strong recommendations for this paper. It reports a lot of data that I for one think are interesting, but, in my view, have a weaker chronological significance then portrayed here. If this paper were published in its present form it would be fine -- readers can clearly obtain and understand the context of the data, even if the data are surrounded by a lot of discussion that is not strongly related to most of the observations. However, I think the paper could be improved by focusing the discussion of the depth-profile data on the process significance and not on the chronology.

In conclusion from the above comments and responses, we provide additional discussion of the depth profile data and permafrost processes in our discussion.

A couple of minor items:

-- The 26/10 diagrams need to say what production rate was used to draw the simple exposure region and isochrons. I take it this is the surface production rate at the core site? The production rates are derived from the surface production rate at Pearse Valley, and lower Wright Valley core sites, respectively. These are outlined in Sect. 3.4. from line 360.

Of course these diagrams look very different if you normalize each sample to the production rate at its respective depth.

We provide additional context at the end of Sect. 5.2 regarding production rates.

-- It would be helpful in 3.2.1 to get a little more information about the ice-cemented soil. The use of 'ice-cemented' tends to indicate that there is only ice in the pore space and it is still clast-supported. Correct? Yes, we provide more information on lines 246 - 248

Also, are the ice wedges in Figure 5 accurately to scale? Yes

As this is a pretty obscure place that is unlikely to be revisited often, it would be great to get a little more detailed description of the sediments.
Additional description is provided in Section 3.1.1

-- Line 360-ish. I am confused by the 'any postdepositional...is unknown...' sentence. Clarify? Maybe what you want to say here is what you know, what you are assuming, and what you are trying to solve for by model fitting.
See updated text in Sect 3.4